# A native prokaryotic voltage-dependent calcium channel with a novel selectivity filter sequence

**Takushi Shimomura[1,2], Yoshiki Yonekawa[3], Hitoshi Nagura[1], Michihiro Tateyama[2], Yoshinori Fujiyoshi[1,4†], Katsumasa Irie[1,3]\***

[1]Cellular and Structural Physiology Institute (CeSPI), Nagoya University, Nagoya, Japan; [2]Division of Biophysics and Neurobiology, National Institute for Physiological Sciences, Okazaki, Japan; [3]Graduate School of Pharmaceutical Sciences, Nagoya University, Nagoya, Japan; [4]CeSPIA Inc, Tokyo, Japan

**Abstract** Voltage-dependent $Ca^{2+}$ channels (Cavs) are indispensable for coupling action potentials with $Ca^{2+}$ signaling in living organisms. The structure of Cavs is similar to that of voltage-dependent $Na^+$ channels (Navs). It is known that prokaryotic Navs can obtain $Ca^{2+}$ selectivity by negative charge mutations of the selectivity filter, but native prokaryotic Cavs had not yet been identified. We report the first identification of a native prokaryotic Cav, CavMr, whose selectivity filter contains a smaller number of negatively charged residues than that of artificial prokaryotic Cavs. A relative mutant whose selectivity filter was replaced with that of CavMr exhibits high $Ca^{2+}$ selectivity. Mutational analyses revealed that the glycine residue of the CavMr selectivity filter is a determinant for $Ca^{2+}$ selectivity. This glycine residue is well conserved among subdomains I and III of eukaryotic Cavs. These findings provide new insight into the $Ca^{2+}$ selectivity mechanism that is conserved from prokaryotes to eukaryotes.

**\*For correspondence:**
kirie@cespi.nagoya-u.ac.jp

**Present address:** [†]Advanced Research Institute, Tokyo Medical and Dental University, Tokyo, Japan

## Introduction

Voltage-dependent $Ca^{2+}$ channels (Cavs), which couple the membrane voltage with $Ca^{2+}$ signaling, regulate some important physiological functions, such as neurotransmission and muscle contraction (*Hille, 2001*). The channel subunits of both mammalian Cavs and mammalian voltage-dependent $Na^+$ channels (Navs) have 24 transmembrane helices (24TM) (*Catterall, 2000*), and comprise four homologous subdomains with six transmembrane helices that correspond to one subunit of homo-tetrameric channels, such as voltage-dependent $K^+$ channels and prokaryotic Navs (BacNavs). Comparison of the sequences of Navs and Cavs indicate that Navs derived from Cavs. Their two pairs of subdomains, domains I and III, and domains II and IV, are evolutionarily close to each other (*Rahman et al., 2014*; *Strong et al., 1993*). Therefore, the 24TM-type of Cavs are thought to have evolved from the single-domain type of Cavs with two domain-duplication events. Although prokaryotes are expected to have such ancestor-like channels, native prokaryotic Cavs have not yet been identified. The lack of ancestor-like prokaryotic Cavs is a missing link in the evolution of voltage-dependent cation channels.

In contrast to the lack of prokaryotic Cavs, many BacNavs have been characterized (*Irie et al., 2010*; *Ito et al., 2004*; *Koishi et al., 2004*; *Lee et al., 2012*; *Nagura et al., 2010*; *Payandeh et al., 2011*; *Ren et al., 2001*; *Shimomura et al., 2016*; *Shimomura et al., 2011*). The simple structure of BacNavs has helped to elucidate the molecular mechanisms of Navs (*Irie et al., 2018*; *Irie et al., 2012*; *Tsai et al., 2013*; *Yue et al., 2002*). In addition, BacNavs have been used as a genetic tool for manipulating neuronal excitability in vivo (*Bando et al., 2016*; *Kamiya et al., 2019*; *Lin et al., 2010*). The acquisition of $Ca^{2+}$ selectivity by BacNavs can be engineered by the introduction of

**eLife digest** Electrical signals in the brain and muscles allow animals – including humans – to think, make memories and move around. Cells generate these signals by enabling charged particles known as ions to pass through the physical barrier that surrounds all cells, the cell membrane, at certain times and in certain locations.

The ions pass through pores made by various channel proteins, which generally have so-called "selectivity filters" that only allow particular types of ions to fit through. For example, the selectivity filters of a family of channels in mammals known as the Cavs only allow calcium ions to pass through. Another family of ion channels in mammals are similar in structure to the Cavs but their selectivity filters only allow sodium ions to pass through instead of calcium ions.

Ion channels are found in all living cells including in bacteria. It is thought that the Cavs and sodium-selective channels may have both evolved from Cav-like channels in an ancient lifeform that was the common ancestor of modern bacteria and animals. Previous studies in bacteria found that modifying the selectivity filters of some sodium-selective channels known as BacNavs allowed calcium ions to pass through the mutant channels instead of sodium ions. However, no Cav channels had been identified in bacteria so far, representing a missing link in the evolutionary history of ion channels.

Shimomura et al. have now found a Cav-like channel in a bacterium known as *Meiothermus ruber*. Like all proteins, ion channels are made from amino acids and comparing the selectivity filter of the *M. ruber* Cav with those of mammalian Cavs and the calcium-selective BacNav mutants from previous studies revealed one amino acid that plays a particularly important role. This amino acid is a glycine that helps select which ions may pass through the pore and is also present in the selectivity filters of many Cavs in mammals.

Together these findings suggest that the Cav channel from *M. ruber* is similar to the mammal Cav channels and may more closely resemble the Cav-like channels thought to have existed in the common ancestor of bacteria and animals. Since other channel proteins from bacteria are useful genetic tools for studies in human and other animal cells, the Cav channel from *M. ruber* has the potential to be used to stimulate calcium signaling in experiments.

several negatively charged amino acids into the selectivity filter (*Tang et al., 2014*; *Yue et al., 2002*). A mutant channel NavAb (a BacNav from *Arcobacter butzleri*) produced in this way, showed high $Ca^{2+}$ selectivity, and the structural basis of $Ca^{2+}$ selectivity has been discussed on the basis of its crystal structures (*Tang et al., 2016*; *Tang et al., 2014*). However, the selectivity filter sequences of CavAb, which were made by mutation of NavAb and contain a large number of aspartates, are quite different from those of the original mammalian Cavs. The evolutionary analysis also indicated that BacNavs acquired sodium selectivity independent from that of 24TM Navs (*Liebeskind et al., 2013*). From these points of view, ancestor-like prokaryotic Cavs could be expected to help us to understand the structural and functional relationship between BacNavs and 24TM channels.

Here, we newly characterized two BacNav homologs, CavMr from *Meiothermus ruber* and NavPp from *Plesiocystis pacifica*. These two channels are evolutionarily distant from the previously reported canonical BacNavs. We confirmed that CavMr has clear $Ca^{2+}$ selectivity, and that NavPp has $Na^+$ selectivity with $Ca^{2+}$-dependent inhibition. The discovery of these channels suggests the possible importance of voltage-regulated $Ca^{2+}$ signaling in prokaryotes and may be a new genetic tool for controlling $Ca^{2+}$ signaling. Furthermore, mutational analyses indicate that the glycine residue of the CavMr selectivity filter is important for $Ca^{2+}$ selectivity. This glycine residue is also well conserved in the selectivity filter of subdomains I and III of mammalian Cavs. On the basis of these observations, we propose that CavMr is an ancestor-like prokaryotic Cav with a $Ca^{2+}$ selectivity mechanism that is different from that in artificial CavAb. Further phylogenetical analyses indicated that CavMr and NavPp homologs form a wide-spread group in prokaryote and archaea, which is different from canonical BacNavs. Therefore, they are expected to advance our understanding of $Ca^{2+}$ recognition and the evolution of voltage-dependent cation channels.

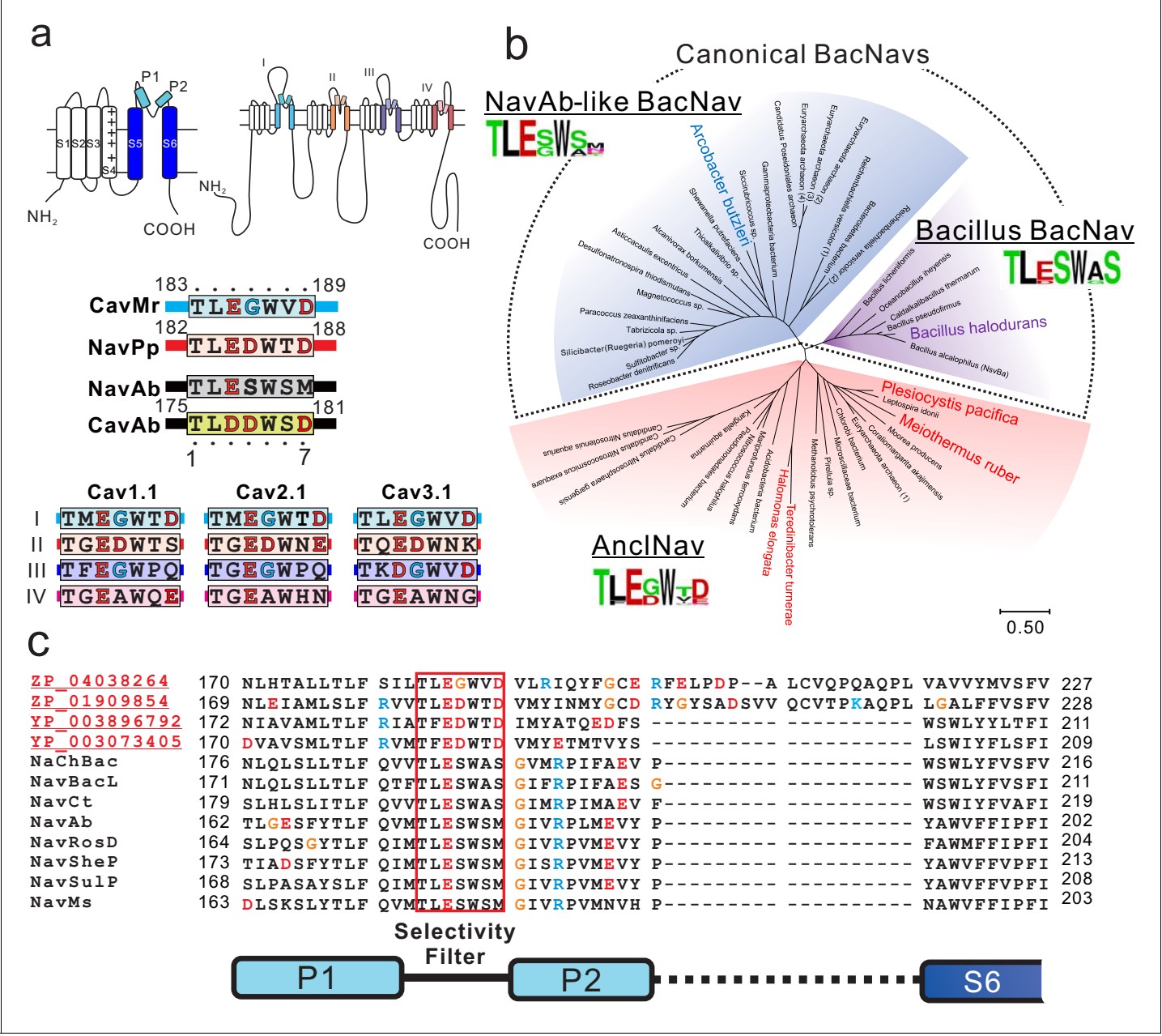

**Figure 1.** Sequence analysis of ancestor-like BacNavs. (a) Schematic secondary structure and selectivity filter sequence of BacNavs and 24TM channels. A cylinder indicates an α-helix. The selectivity filter sequences are indicated using single-letter codes. Negatively charged residues are colored in red. Glycine residues in the position four are colored in cyan. The straight lines indicate the other parts of the pore domain. The selectivity filter sequences of hCav1.1 (UniProt ID: Q13698), hCav2.1 (O00555) and hCav3.1 (O43497) were used. (b) Phylogenetic tree of canonical BacNavs and ancestor-like BacNavs (AncINavs). The MUSCLE program was used to align the multiple protein sequences of the channels (*Figure 1—source data 1*). The phylogenetic tree was generated using MEGA X. The branch lengths are proportional to the sequence divergence, with the scale bar corresponding to 0.5 substitutions per amino acid position. Three phylogenetically distinct groups are shown in different background colors (purple, Bacillus BacNavs; blue, NavAb-like BacNavs; red, AncINavs). Four homologs with the taxon name colored in red in the AncINav group were cloned and expressed to check the channel activity. Two of those, which are shown in larger and bold text, generated the detectable currents. The appearance frequency of amino acids in each of the selectivity filter sequences is shown under the respective group names. (c) Alignment of the deduced amino-acid sequences of the P1 helix to P2 helix domain of novel cloned homologs of AncINavs with well characterized BacNavs.

The online version of this article includes the following source data and figure supplement(s) for figure 1:

**Source data 1.** Amino-acid sequences used for making phylogenetic tree.
**Figure supplement 1.** Distribution of the bacterial phylum and archaea in the phylogenetic tree of BacNavs.
**Figure supplement 2.** Phylogenetic analysis of BacNavs with eukaryotic voltage-gated cation channels.
*Figure 1 continued on next page*

*Figure 1 continued*

**Figure supplement 2—source data 1.** Amino-acid sequences used for making phylogenetic tree.

## Results

### Identification of a prokaryotic channel with Ca$^{2+}$ permeability

We searched for the primary sequences of candidate prokaryotic Cavs in the GenBank database. In mammalian and prokaryotic Navs and Cavs, a larger number of negative charges in the filter increases Ca$^{2+}$ selectivity (*Figure 1a*) (*Heinemann et al., 1992*; *Tang et al., 2014*; *Yue et al., 2002*). Several BLAST search rounds using the pore regions (S5–S6) of NaChBac (or NavBh; a BacNav from *Bacillus halodurans*) as templates revealed a series of candidate prokaryotic Cavs whose selectivity filters are similar to the 'TLESW' motif, but which contain more negatively charged residues like the filter sequence of CavAb (*Figure 1a*). Phylogenetic analysis of these channel genes revealed that they apparently belong to a different branch of the tree than that of canonical Bac-Navs, namely a *Bacillus* group and a NavAb-like group (*Figure 1b*; *Figure 1—source data 1*). The selectivity filter sequences of these channels are similar to that of the ancestral BacNav channel predicted previously (*Liebeskind et al., 2013*). Therefore, we named these channels an̲c̲e̲s̲t̲o̲r̲-l̲i̲k̲e̲ Bac-Navs (AnclNavs).

AnclNavs are widely distributed in multiple bacterial phyla and even in archaea (*Figure 1—figure supplement 1*; *Figure 1—source data 1*). In some cases, one phylum, such as proteobacteria, contains both a NavAb-like BacNav and an AnclNav gene. Even in those cases, the NavAb-like BacNav and the AnclNav were included separately in their respective groups. The bacillus BacNav group is a different group located phylogenetically between the NavAb-like and the AnclNav groups. In addition, the firmicutes phylum, which includes Bacillus species, contains neither homologs of NavAb-like BacNavs nor homologs of AnclNavs. These observations suggest that our identified candidate prokaryotic Cavs, AnclNavs, are homologs rather than orthologs of canonical BacNavs and compose a distinct group. In addition, analyses that include some eukaryotic channels, such as each subdomain of 24TM-type of Navs/Cavs, CatSper and EukCatA, a group of eukaryotic non-selective homotetrameric channels, put AnclNavs closest to a *Bacillus* group (*Figure 1—figure supplement 2*; *Figure 1—figure supplement 2—source data 1*).

We identified four AnclNavs genes and measured their channel activity: ZP_04038264 from *M. ruber*, ZP_01909854 from *P. pacifica*, YP 003896792_from *Halomonas elongata*, and YP_003073405 from *Teredinibacter turnerae* (*Figure 1b and c*). When attempting to express prokaryotic channels transgenetically, insect cells are often better than mammalian cells for generating large current amplitudes (*Irie et al., 2018*). We therefore transfected Sf9 cells with these four channel genes and measured the resulting whole-cell currents. The cells that were transfected with genes from *M. ruber* showed currents in response to a depolarizing stimulus from a −140 mV holding potential (*Figure 2a*). To estimate the Ca$^{2+}$ permeability, we measured their current-voltage relationships. The *M. ruber* channel clearly had larger currents in the high-Ca$^{2+}$ solution than in the high-Na$^+$ solution, and no obvious outward current was observed in a high-Ca$^{2+}$ bath solution, even at very high membrane potential (100 mV) (*Figure 2a and b*; *Figure 2—source data 1*). These current-voltage relationships suggest that the *M. ruber* channel has a preference for Ca$^{2+}$, and that other cations inside the cells (sodium and cesium) hardly permeated the activated channel. Therefore, the newly identified channel from *M. ruber* is abbreviated as CavMr, based on its ion selectivity and species name. We evaluated the voltage-dependent activation of CavMr by measuring deactivation tail currents (*Figure 2c*). A Boltzmann fit of the averaged activation curve yielded an activating potential of 50% activation (V$_{1/2}$) of −51.7 ± 1.1 mV (*Figure 2d*; *Figure 2—source data 2*).

To compare clearly the positions of the residues in the selectivity filter in each channel, we renumbered the seven residues comprising the selectivity filter. For example, the seven residues of the CavMr selectivity filter are 183-TLEGWVD-189, and thus Thr183 and Asp189 were renumbered as Thr1 and Asp7 (*Figure 1a*). Notably, the amino acid sequence of the selectivity filter in CavMr is similar to the conserved features of domains I/III in mammalian Cavs, with a glycine at position 4 and a polar or negatively charged residue at position 7, which are not observed in the canonical BacNav

family. In addition, the CavMr selectivity filter sequence is quite similar to that of the human Cav sub-domain I, or even the same as Cav3.1 and 3.2 (*Figure 1a*).

In the following experiments, to evaluate the reversal potential for the ion selectivity analysis accurately, we introduced a single mutation that resulted in large and long-lasting channel currents. T220A and G229A mutations in NaChBac led to slower inactivation and provided a larger current, indicating suppression of the transition to the inactivated state (*Irie et al., 2010*; *Shimomura et al., 2016*). We introduced a G240A mutation to CavMr, corresponding to the NaCh-Bac mutations of G229A. Th is mutant channels stably showed larger and more measurable currents than the wild-type channel, even after they were administrated multiple depolarizing stimuli (*Figure 2e and f*).

## CavMr has high Ca²⁺ selectivity over Na⁺

We accurately quantified the selectivity of CavMr for Na⁺ and Ca²⁺ ($P_{Ca}/P_{Na}$) by measuring the reversal potential ($E_{rev}$) under bi-ionic conditions, in which the Ca²⁺ concentration in the bath solution was changed to 4, 10, 20, and 40 mM while the intracellular Na⁺ concentration was held constant at 150 mM (*Figure 3a and b*; *Figure 3—figure supplement 1a–d*; *Figure 3—source data 1*). The plot of the reversal potentials as a function of $[Ca^{2+}]_{out}$ had a slope of 39.89 ± 3.31 mV/decade (*n* = 4). It

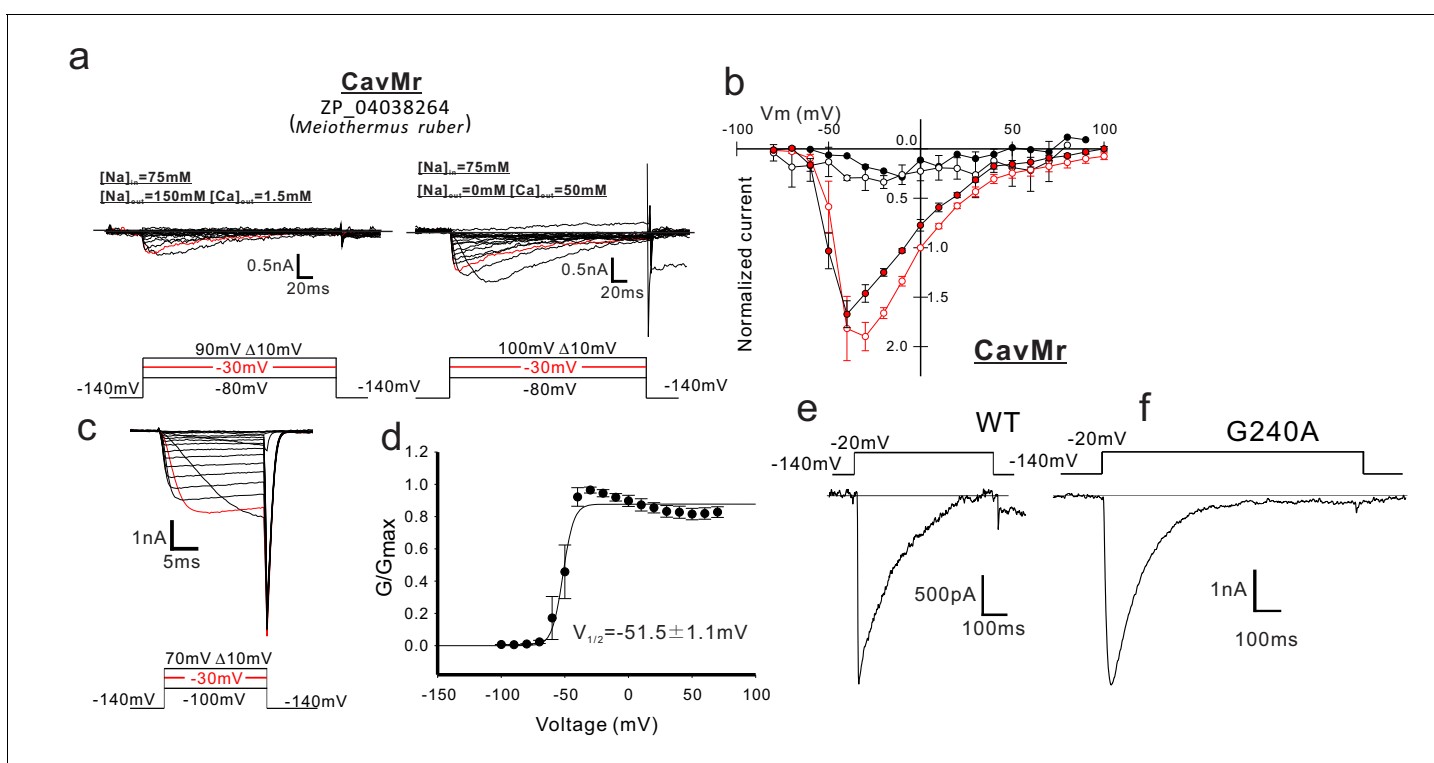

**Figure 2.** Functional expression of CavMr in SF-9 insect cells. (**a**) Representative current traces used to describe the current-voltage relationships of CavMr in SF9 cells. The horizontal lines are superimposed to indicate the zero-current level in the representative current traces. Currents were generated in the bath solutions containing high Na⁺ (left) and high Ca²⁺ (right), by a series of step-pulses (shown at the bottom of the panel). (**b**) Current-voltage relationships of CavMr measured in the different bath solutions [filled black, 150 mM NaCl (*n* = 4); open black, 75 mM NaCl and 75 mM NMDG-HCl (*n* = 4); open red, 75 mM NaCl and 50 mM CaCl₂ (*n* = 7); filled red, 50 mM CaCl₂ and 75 mM NMDG-HCl (*n* = 6)] (*Figure 2—source data 1*). Currents of CavMr were normalized to that invoked by 0 mV depolarization stimuli under 75 mM NaCl and 50 mM CaCl₂ bath solution. (**c**) Deactivation tail currents of CavMr. After prepulses of varying depolarization (bottom), tail currents were measured at −140 mV. (**d**) G/Gmax curve of CavMr generated by tail currents (*n* = 6) (*Figure 2—source data 2*). (**e, f**) Whole-cell currents in CavMr wild type [WT; (**e**) ] and a G240A mutant (**f**) when a pulse of −20 mV was given for 500 ms and 1 s, respectively, in a high Ca²⁺ bath solution.

The online version of this article includes the following source data for figure 2:

**Source data 1.** The values of the currents generated by each voltage stimulation.
**Source data 2.** The values of G/Gmax of CavMr derived from the tail currents generated by each voltage stimulation.

was higher than the Nernst prediction for a divalent cation (29 mV). We think that this deviation came from incorporation of the measurements made with the 40 mM $Ca^{2+}$ bath solution, in which no obvious outward current was observed (*Figure 3—figure supplement 1d*), resulting in a higher value of $E_{rev}$. The well-fitted plot of the reversal potentials in 4, 10 and 20 mM $Ca^{2+}$ bath solutions as a function of $[Ca^{2+}]_{out}$ did indeed have a slope of 32.38 ± 4.10 mV/decade ($n$ = 4) (*Figure 3b* dashed line), a value close to the Nernst equation for $Ca^{2+}$ (*Figure 3b*), and indicated that CavMr had a $P_{Ca}/P_{Na}$ of 218 ± 38 (*Figure 3e*, *Table 1*; *Figure 3—source data 2*). This high $P_{Ca}/P_{Na}$ value is comparable to that of CavAb. Among several species of cations, including $Sr^{2+}$, $K^+$, and $Cs^+$, $Ca^{2+}$ had the highest permeability relative to $Na^+$ (*Figure 3c,d and f*, *Table 1*; *Figure 3—source data 3*). On the basis of these results, CavMr was confirmed to be a native prokaryotic Cav with high $Ca^{2+}$ selectivity. We also investigated whether CavMr shows the typical anomalous mole fraction effect (*Almers and McCleskey, 1984*) and the non-monotonic mole fraction effect observed in NaChBac

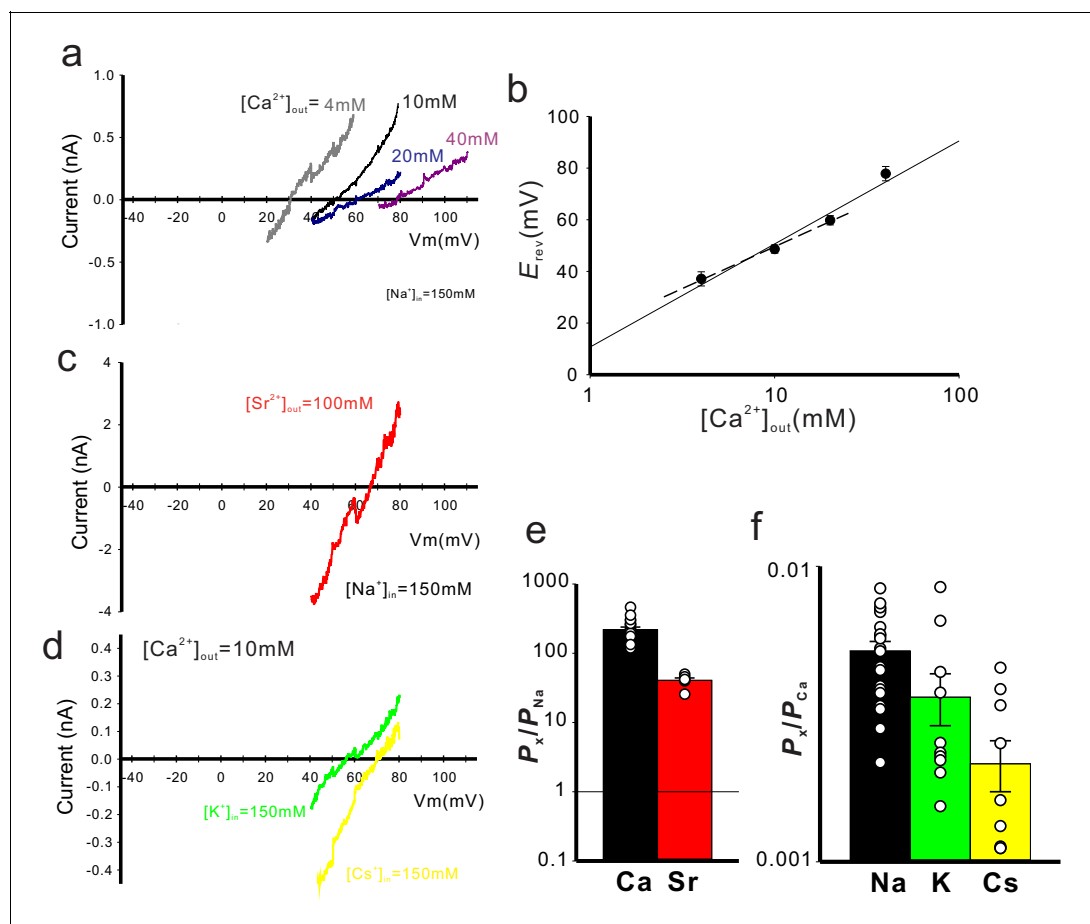

**Figure 3.** Cation selectivity of CavMr. (a) Current-voltage relationship plot generated by ramp pulses in various $[Ca^{2+}]_{out}$ and 150 mM $[Na^+]_{in}$. (b) The plot of the reversal potential to $[Ca^{2+}]_{out}$. Each value was obtained using the ramp pulse protocol shown in panel (a) (*Figure 3—source data 1*). The relationship was fitted by a line with the slope of 39.89 ± 3.31 mV per decade ($n$ = 7). (c) Current-voltage relationship plot generated by ramp pulses in 100 mM $[Sr^{2+}]_{out}$ and 150 mM $[Na^+]_{in}$. (d) Current-voltage relationship plots generated by ramp pulses in 10 mM $[Ca^{2+}]_{out}$ and 150 mM $[K^+]_{in}$ or $[Cs^+]_{in}$. (e) The relative permeability of $Ca^{2+}$ or $Sr^{2+}$ to $Na^+$ in CavMr, calculated from the reversal potentials that were obtained by the ramp pulses shown in *Figure 3—figure supplement 1* (*Figure 3—source data 2*). (f) The relative permeability of each monovalent cation to $Ca^{2+}$ in CavMr, derived from the data shown in *Figure 3—figure supplement 1* (*Figure 3—source data 3*).

The online version of this article includes the following source data and figure supplement(s) for figure 3:

**Source data 1.** The reversal potentials to each extracellular $Ca^{2+}$ concentration.

**Source data 2.** The values of relative permeability.

**Source data 3.** The values of relative permeability.

**Figure supplement 1.** Representative current traces of CavMr generated by the ramp protocol.

**Table 1.** Relative permeability of CavMr and NavPp.
All values are indicated as mean ± S.E.

| | $P_{Ca}/P_{Na}$ | $_{Sr}/P_{Na}$ | $P_K/P_{Na}$ | $P_{Cs}/P_{Na}$ |
|---|---|---|---|---|
| CavMr G240A | 218 ± 38 | 40.6 ± 3.4 | 0.0036 ± 0.00072[a] | 0.0021 ± 0.00042[b] |
| | (n = 20) | (n = 6) | (n = 10) | (n = 10) |
| Pp | 13.8 ± 2.0 | 24.5 ± 0.3 | 0.95 ± 0.04 | 0.57 ± 0.05 |
| | (n = 7) | (n = 5) | (n = 4) | (n = 3) |
| G4D | 7.73 ± 2.24 | 18.6 ± 6.1 | 1.20 ± 0.28 | 0.87 ± 0.21 |
| | (n = 11) | (n = 4) | (n = 4) | (n = 4) |
| G4S | 11.9 ± 1.5 | 4.23 ± 0.27 | 1.54 ± 0.12 | 2.02 ± 0.48 |
| | (n = 5) | (n = 5) | (n = 5) | (n = 3) |
| V6T | 40.1 ± 9.7 | 13.3 ± 2.5 | 0.69 ± 0.26 | 0.54 ± 0.60 |
| | (n = 5) | (n = 5) | (n = 3) | (n = 3) |
| D7M | 144 ± 12 | 20.7 ± 2.7 | N.D. | N.D. |
| | (n = 5) | (n = 5) | | |
| NavPp T232A | 0.308 ± 0.028 | 0.38 ± 0.027 | 0.16 ± 0.026 | 0.0052 ± 0.0006 |
| | (n = 18) | (n = 9) | (n = 9) | (n = 7) |
| Mr | 215 ± 33 | 86.3 ± 12.2 | 0.0045 ± 0.00072[a] | 0.0135 ± 0.0039[b] |
| | (n = 7) | (n = 4) | (n = 4) | (n = 8) |
| D4G | 41.4 ± 6.7 | 8.85 ± 0.95 | 0.81 ± 0.11 | 0.56 ± 0.05 |
| | (n = 10) | (n = 4) | (n = 3) | (n = 4) |
| T6V | 1.72 ± 0.19 | 33.9 ± 5.0 | 0.99 ± 0.03 | 0.84 ± 0.02 |
| | (n = 10) | (n = 8) | (n = 4) | (n = 4) |

[a] Because of high $Ca^{2+}$ selectivity, $P_K/P_{Ca}$ are indicated.
[b] Because of high $Ca^{2+}$ selectivity, $P_{Cs}/P_{Ca}$ are indicated.

(*Finol-Urdaneta et al., 2014*). CavMr did not allow $Na^+$ permeation under $Ca^{2+}$-free (0 mM $CaCl_2$ and 1 mM EGTA) conditions (*Figure 4a and b*; *Figure 4—source data 1*). Also, in contrast to the recording of NaChBac currents in a solution containing $Na^+$ and $K^+$, CavMr had an apparently monotonic current increase depending on the $Ca^{2+}$ mole fraction to $Na^+$ (*Figure 4c and d*; *Figure 4—source data 2*).

Studies of an artificial Cav, CavAb, revealed that $Ca^{2+}$ selectivity depends on the presence of a large number of aspartates in the filter sequence (*Tang et al., 2014*). The high $Ca^{2+}$ selectivity in CavMr was unexpected because the filter sequence contained only one aspartate residue (*Figure 1c*). Furthermore, CavMr-D7M, which has only one negatively charged residue in the selectivity filter 'TLEGWVM', still had high $Ca^{2+}$ selectivity, comparable to that of wild-type CavMr ($P_{Ca}/P_{Na}$ = 144 ± 12; *Figure 4e-g* and *Table 1*; *Figure 4—source data 3*). These findings indicate that CavMr and artificial CavAb have different $Ca^{2+}$-selection mechanisms.

## NavPp is permeable to $Na^+$ and is blocked by extracellular $Ca^{2+}$

The currents derived from the *P. pacifica* channel became large with increases in the bath $Na^+$ concentration and significantly decreased when the $Na^+$ solution was replaced with a high $Ca^{2+}$ solution (*Figure 5a and b*; *Figure 5—source data 1*). Because the reversal potential fit well to the $Na^+$ equilibrium potential in the high-$Na^+$ solution (*Figure 5b*), we abbreviated this channel as NavPp on the basis of its ion selectivity and species name. Interestingly, NavPp, despite having one more aspartate in the selectivity filter than CavMr, exhibited larger currents in $Na^+$ solutions than in $Ca^{2+}$ solutions (*Figure 1c* and *Figure 5a*). This observation indicates that $Ca^{2+}$ selection is not achieved simply by increasing negative charges in the filter sequence of the AncINav group, which again suggests the existence of an alternative ion-selectivity mechanism.

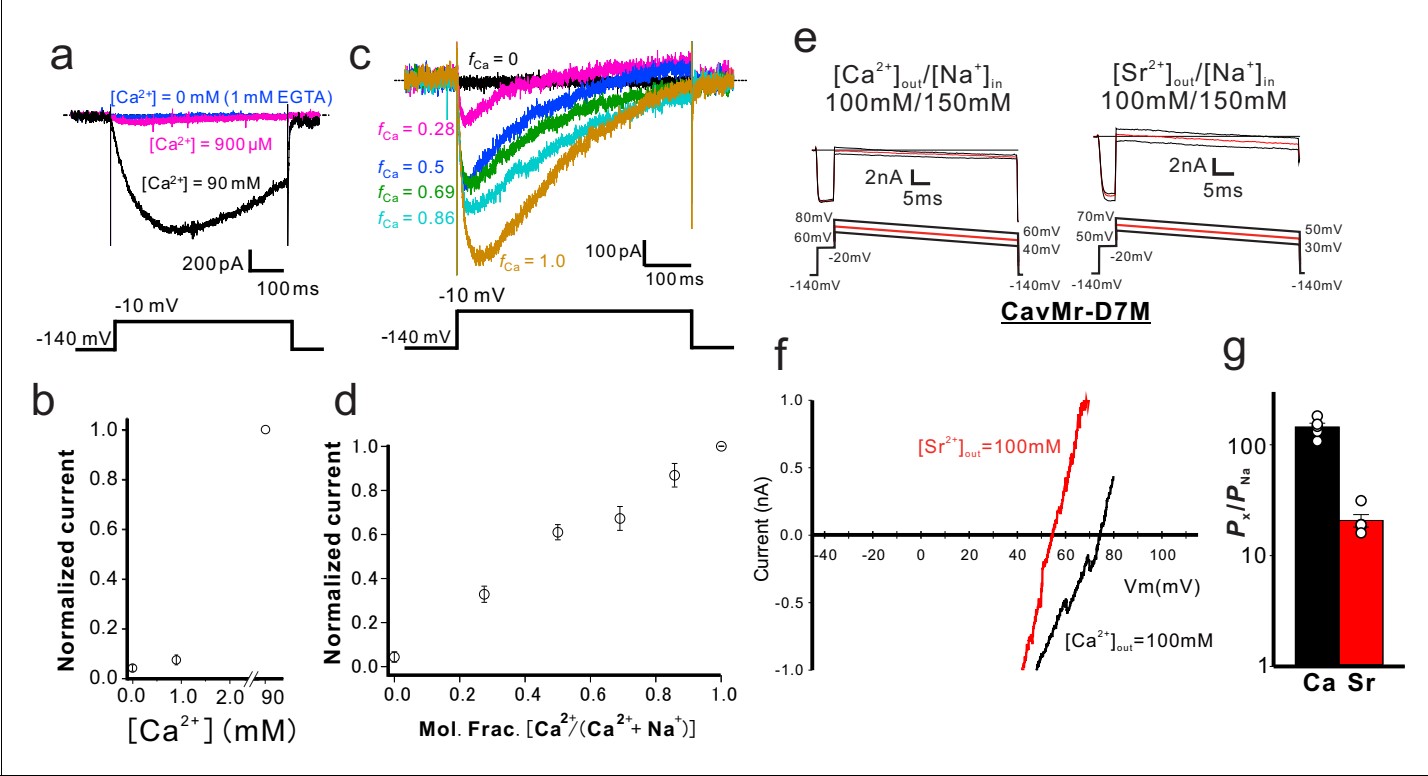

**Figure 4.** Characterization of the selectivity filter of CavMr. (a) Examination of anomalous mole fraction effects in CavMr. CavMr currents were recorded in a bath solution containing the following ratios of $Na^+$ and $Ca^{2+}$ ($[Na^+]$:$[Ca^{2+}]$) — 0:90, 133.7:0.9 and 135:0 (mM), respectively. The 0 mM $Ca^{2+}$ solution also contains 1 mM EGTA. (b) Plot of the normalized current amplitude of CavMr obtained from panel (a) ($n$ = 3) (*Figure 4—source data 1*). (c) Representative current traces of CavMr under different mole fractions of $Ca^{2+}$. $f_{Ca}$ indicates $[Ca^{2+}]_{out}$ / ($[Ca^{2+}]_{out}$ + $[Na^+]_{out}$). (d) Plot of the normalized current amplitude to each mole fraction, as measured in c ($n$ = 5) (*Figure 4—source data 2*). (e) For the evaluation of the relative permeability of $Ca^{2+}$ and $Sr^{2+}$ to $Na^+$ of CavMr-D7M, $Ca^{2+}$ solution [100 mM $CaCl_2$, 10 mM HEPES (pH 7.4 adjusted with $Ca(OH)_2$) and 10 mM glucose] and $Sr^{2+}$ solution [100 mM $SrCl_2$, 10 mM HEPES (pH 7.4 adjusted by $Sr[OH]_2$) and 10 mM glucose] were used as bath solutions. High-$Na^+$ pipette solution [115 mM NaF, 35 mM NaCl, 10 mM EGTA, and 10 mM HEPES (pH 7.4 adjusted by NaOH)] was used. Currents were generated by the step pulse of −20 mV from −140 mV holding potential, followed by ramp pulses with different voltage values. The time courses of the change of membrane potentials are shown at the bottom of each current traces. (f) Current-voltage relationship plots generated by ramp pulses in 150 mM $[Na^+]_{in}$ and 100 mM $[Ca^{2+}]_{out}$ or $[Sr^{2+}]_{out}$. (g) The relative permeability of divalent cations to $Na^+$ in CavMr-D7M, whose position 7 residue in the selectivity filter was neutralized by the corresponding residue of NavAb (*Figure 4—source data 3*).

The online version of this article includes the following source data for figure 4:

**Source data 1.** The values of the normalized current amplitude of CavMr.

**Source data 2.** The values of the normalized current amplitude to each mole fraction.

**Source data 3.** The values of relative permeability.

Recordings in bath solution containing both $Na^+$ and $Ca^{2+}$ demonstrated that the increment of the extracellular $Ca^{2+}$ decreased the current in NavPp and led to a positive shift in the voltage dependence, suggesting that a higher concentration of $Ca^{2+}$ inhibited the gating and ionic permeation of NavPp (*Figure 5b*). We tried to measure the voltage-dependent activation of NavPp, but the wild-type channel showed very fast deactivation and no tail current was observed (*Figure 5c*). By introducing a T232A mutation, corresponding to the NaChBac mutations of T220A (*Shimomura et al., 2016*), we were able to observe the tail current at −60 mV (*Figure 5d and e*). A Boltzmann fit of the averaged activation curve of NavPp T232A yielded a $V_{1/2}$ of −17.11 ± 1.8 mV (*Figure 5f*; *Figure 5—source data 2*). To characterize the effect of extracellular $Ca^{2+}$ on the NavPp channel, the voltage dependence of activation of NavPp was measured under various bath $Ca^{2+}$ concentrations (*Figure 6a–c* and *Figure 6—figure supplement 1a*; *Figure 6—source data 1* and *Figure 6—source data 2*). The increments of the extracellular $Ca^{2+}$ raised the value of the reversal potential, indicating that extracellular calcium ions can also permeate NavPp (*Figure 6b*). However,

in higher extracellular $Ca^{2+}$ concentrations, the current amplitude of NavPp became smaller, even in the voltage at which NavPp opens fully, and the voltage dependence of the activation shifted more positively (*Figure 6b–c*). These results indicated that calcium ions can permeate NavPp but disturb the gating and ionic permeation of NavPp.

We then compared the relative permeability of various cations with that of $Na^+$ in NavPp. The reversal potential was obtained under an extracellular solution containing $Na^+$ ions, despite a partial $Ca^{2+}$- or $Sr^{2+}$-induced block (*Figure 6—figure supplement 1b and c*). The selectivity of NavPp was higher for $Na^+$ than for $Ca^{2+}$, $Sr^{2+}$, $K^+$, or $Cs^+$ (*Figure 6d and e*; *Figure 6—source data 3*). The $P_{Ca}/P_{Na}$ was $0.308 \pm 0.028$ in a bath solution containing both $Ca^{2+}$ and $Na^+$, suggesting that a larger fraction of $Ca^{2+}$ is allowed to permeate with outside $Na^+$ ions through NavPp than through canonical BacNavs. Similar to $Ca^{2+}$, $Sr^{2+}$ also blocked the NavPp current, but may also permeate the channel along with $Na^+$ ions (*Figure 6—figure supplement 1c*). These findings demonstrate a unique feature of NavPp, a low-affinity $Ca^{2+}$ block, which is not reported in canonical BacNavs.

Interestingly, the filter sequence of NavPp, 'TLEDWTD', has three negatively charged residues, similar to the filter sequences of the artificial $Ca^{2+}$-selective BacNav mutants (the 'TLEDWSD' mutant

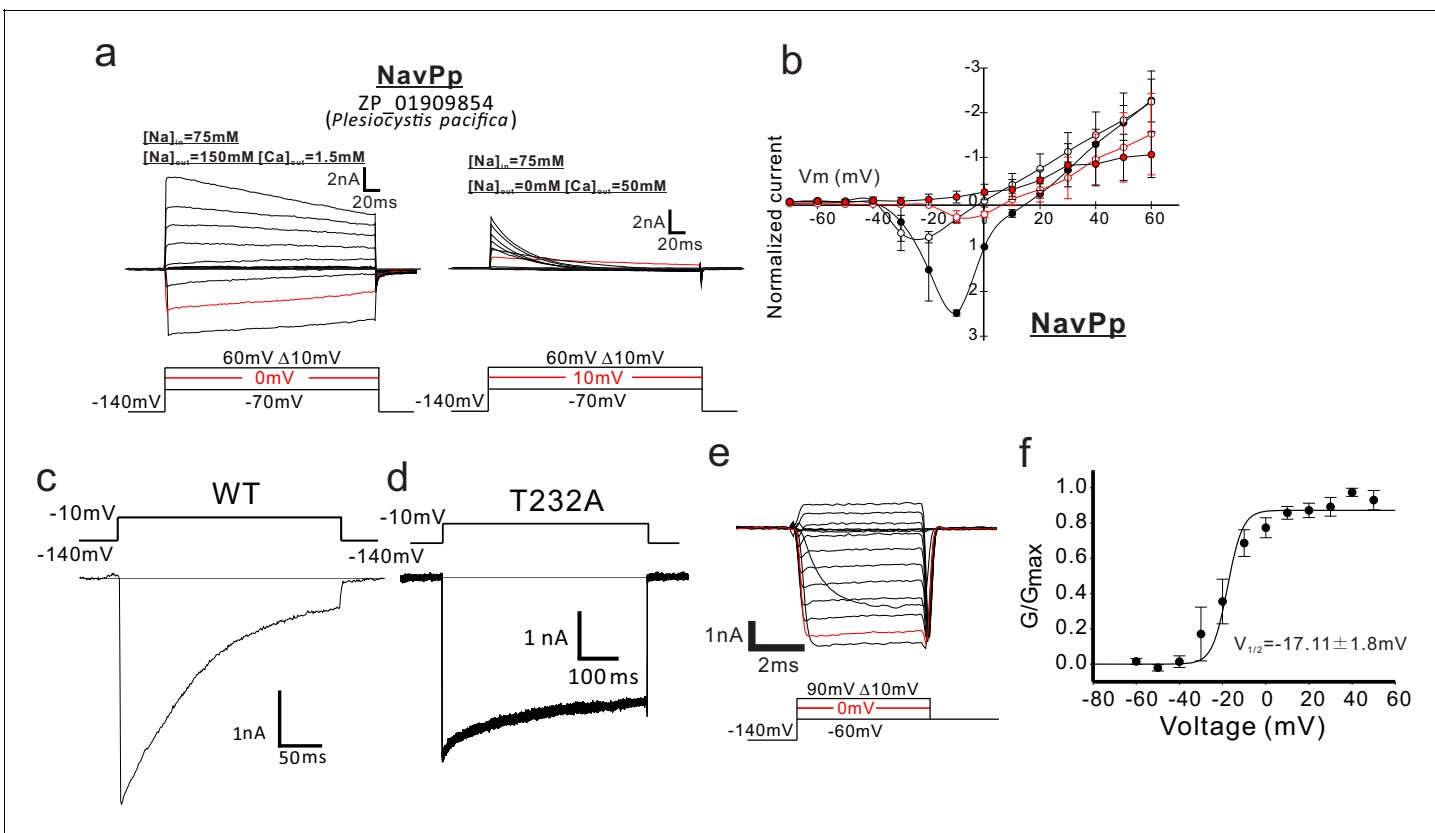

**Figure 5.** Functional expression of NavPp in SF-9 insect cells. (a) Representative current traces used to obtain the current-voltage relationships of NavPp in SF9 cells. The horizontal lines are superimposed to indicate the zero-current level in the representative current traces. Currents were generated, in bath solutions containing high $Na^+$ (left) and high $Ca^{2+}$ (right), by a series of step-pulses shown at the bottom of the panel. (b) Current-voltage relationships of NavPp measured in the different bath solutions [filled black, 150 mM NaCl (*n* = 8), open black, 75 mM NaCl and 75 mM NMDG-HCl (*n* = 8); open red, 75 mM NaCl and 50 mM CaCl₂ (*n* = 6); filled red, 50 mM CaCl₂ and 75 mM NMDG-HCl (*n* = 8)] (*Figure 5—source data 1*). Currents of NavPp were normalized to that induced by 0 mV depolarization stimuli in a 150 mM NaCl bath solution. (c, d) Whole-cell recordings of wild-type NavPp [WT; (c)] and the NavPp T232A mutant (d) when a pulse of −10 mV was given for 250 ms and 500 ms in a high-Na⁺ bath solution, respectively. (e) Deactivation tail currents of NavPp T232A. After prepulses of varying depolarizing currents (bottom), tail currents were measured at −60 mV. (f) G/G_max curve for NavPp T232A derived from the tail currents (*n* = 4) (*Figure 5—source data 2*).

The online version of this article includes the following source data for figure 5:

**Source data 1.** The values of the currents generated by each voltage stimulation.

**Source data 2.** The values of G/G_max of NavPp T232A derived from the tail currents generated by each voltage stimulation.

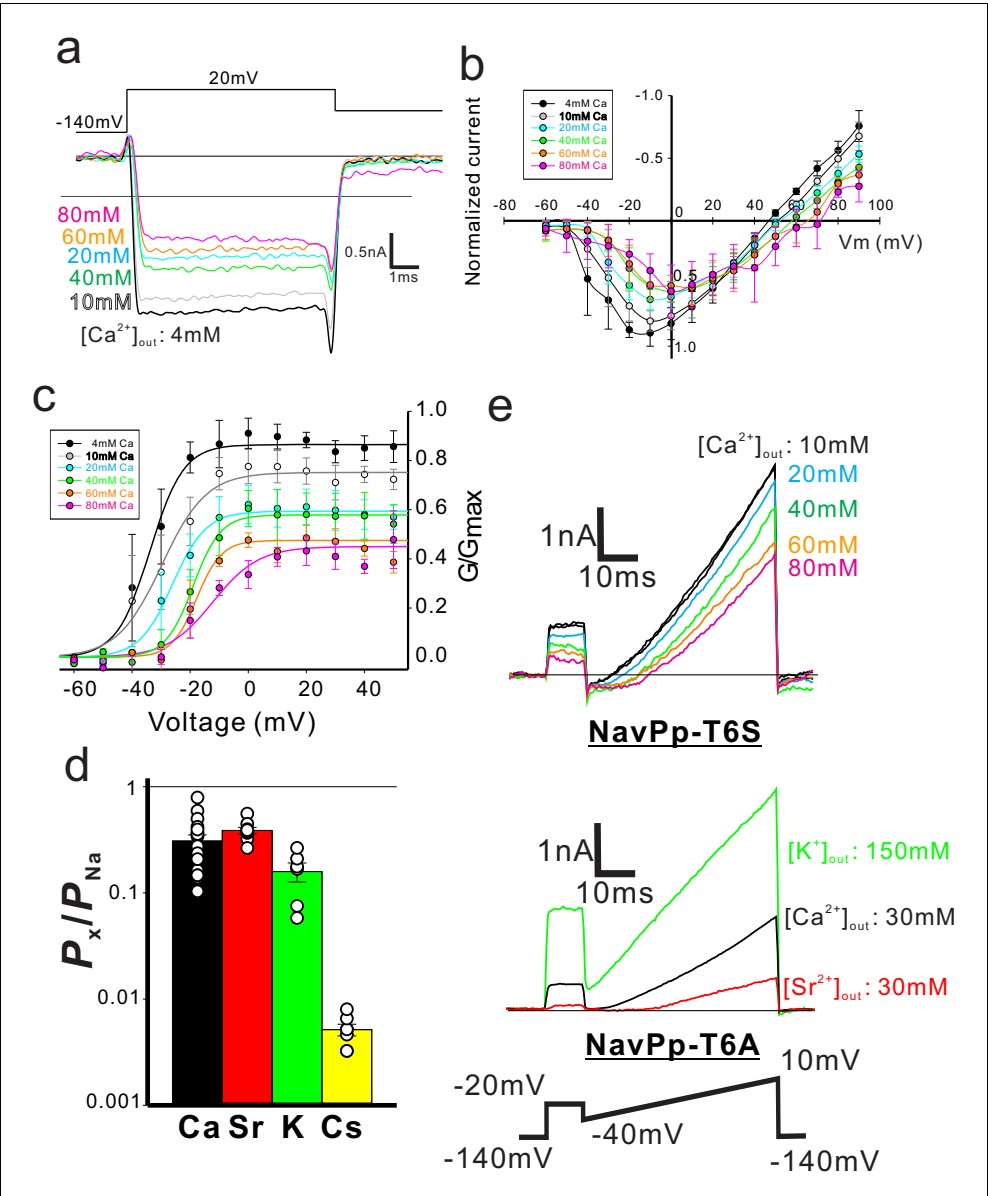

**Figure 6.** Characterization of the selectivity filter of NavPp. (**a**) Representative current traces for NavPp T232A generated by +20 mV stimulation pulses in various extracellular $Ca^{2+}$ concentration solutions. (**b**) Current-voltage relationships of NavPp measured in various extracellular $Ca^{2+}$ concentration solutions (*n* = 3) (***Figure 6—source data 1***). All values were normalized by the peak current amplitude in the 4 mM extracellular $Ca^{2+}$ condition. (**c**) G/ $G_{max}$ curve for NavPp T232A derived from the tail currents in various extracellular $Ca^{2+}$ concentration solutions (*n* = 4) (***Figure 6—source data 2***). The maximum tail current amplitude in the 4 mM extracellular calcium condition was used as $G_{max}$. (**d**) The permeability of different cation species relative to $Na^+$ permeability in NavPp, calculated from the reversal potential that was obtained from the current traces of ***Figure 6—figure supplement 1b–e*** (***Figure 6—source data 3***). (**e**) The extracellular-calcium-inhibition in the single-point mutants of NavPp. The selectivity filter of NavPp was changed to the $Ca^{2+}$-selective canonical-BacNavs mutants (T6S; TLEDWSD and T6A; TLEDWAD).

The online version of this article includes the following source data and figure supplement(s) for figure 6:

**Source data 1.** The values of the currents generated by each voltage stimulation under each extracellular $Ca^{2+}$ concentration.

**Source data 2.** The values of G/Gmax of NavPp T232A derived from the tail currents generated by each voltage stimulation under each extracellular $Ca^{2+}$ concentration.

**Source data 3.** The values of relative permeability.

*Figure 6 continued on next page*

*Figure 6 continued*

**Figure supplement 1.** Representative current traces of NavPp generated by the step-up pulses and the ramp protocol.

of NavAb and the 'TLEDWAD' mutant of NaChBac) (*Tang et al., 2014*; *Yue et al., 2002*). NavPp does not show high $Ca^{2+}$ permeability, but rather a $Ca^{2+}$ block. We also investigated NavPp mutants that have the same filter sequences as the artificial Cavs. NavPp-T6S 'TLEDWSD' exhibited $Ca^{2+}$-blocked currents similar to those exhibited by wild-type NavPp (*Figure 6e*: upper). Further, NavPp-

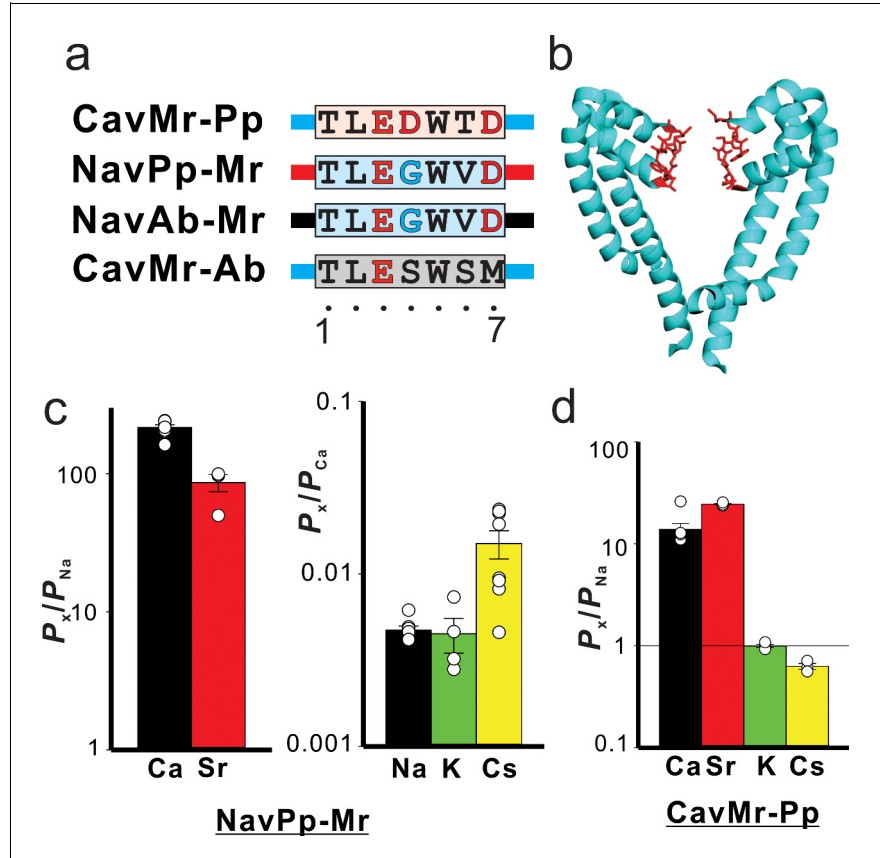

**Figure 7.** The cation selectivity of the channel mutants in which the selectivity filter is swapped between CavMr and NavPp. (a) Amino acid sequences of the selectivity filter in the swapped mutants, CavMr-Pp, Nav-Pp, NavAb-Mr, and CavMr-Ab. The selectivity filter sequences of CavMr, NavPp and NavAb are indicated using single-letter codes with cyan, red, and gray shade, respectively. Negatively charged residues are colored in red. Glycine residues are colored in cyan. The straight lines of cyan, red, and black indicate the other part of pore domain of CavMr, NavPp, and NavAb, respectively. (b) Pore domains of crystal structure of NavAb (PDB code:5YUA). The selectivity filter, which corresponds to the sequences shown in panel (a), was indicated in red. (c) The relative permeability of divalent cations to $Na^+$ (left) and that of monovalent cations to $Ca^{2+}$ (right) in NavPp-Mr (*Figure 7—source data 1*). (d) The relative permeability of different cation species to $Na^+$ in CavMr-Pp (*Figure 7—source data 2*).

The online version of this article includes the following source data and figure supplement(s) for figure 7:

**Source data 1.** The values of relative permeability.
**Source data 2.** The values of relative permeability.
**Figure supplement 1.** Current-voltage relationship plots and representative current traces of the ramp pulse of the CavMr-Pp selectivity-filter-swapped mutants.
**Figure supplement 2.** Current-voltage relationship plots and representative current traces for the ramp pulse of NavPp-Mr selectivity-filter-swapped mutants.

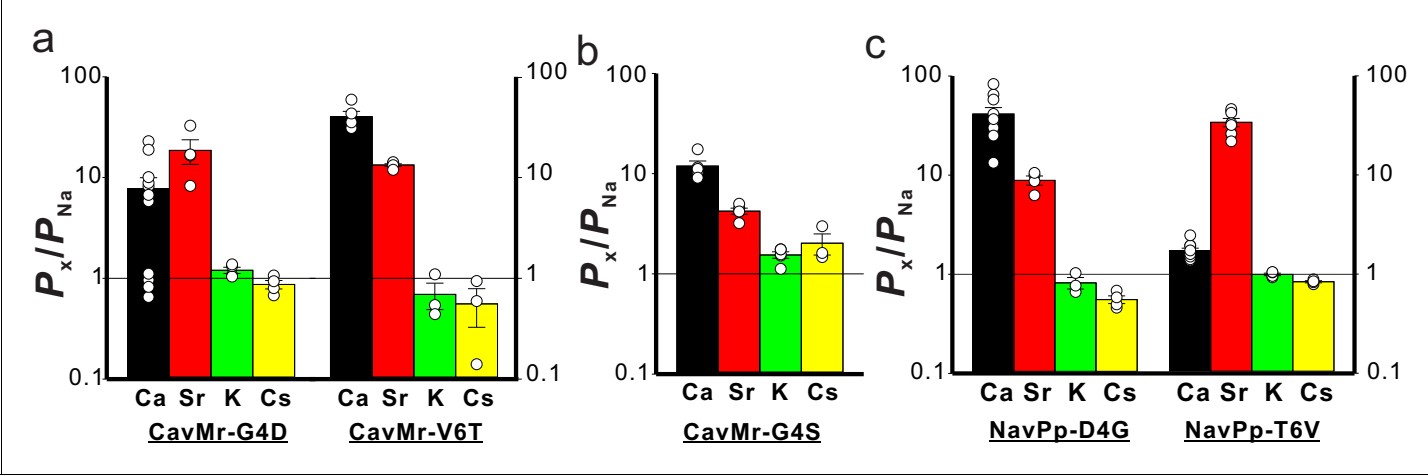

**Figure 8.** The single-point mutations that cause loss and acquistion of $Ca^{2+}$ selectivity in CavMr and NavPp, respectively. (a) The permeability of each cation species relative to $Na^+$ permeability in the single-point mutants of CavMr. The selectivity filter of CavMr was changed to the corresponding residues of NavPp at position 4 (G4D) or position 6 (V6T) (**Figure 8—source data 1**). (b) The permeability of each cation species relative to $Na^+$ permeability in the G4S mutant of CavMr, whose position 4 residue of the selectivity filter was mutated to the corresponding residue of canonical BacNavs (**Figure 7—source data 1**). (c) The permeability of each cation species relative to $Na^+$ permeability in the single-point mutants of NavPp. The selectivity filter of NavPp was changed by swapping in the corresponding residues of CavMr at position 4 (D4G) or position 6 (T6V) (**Figure 8—source data 3**).

The online version of this article includes the following source data and figure supplement(s) for figure 8:

**Source data 1.** The values of relative permeability.
**Source data 2.** The values of relative permeability.
**Source data 3.** The values of relative permeability.
**Figure supplement 1.** Current-voltage relationship plots and representative current traces for the ramp pulse of CavMr-G4D selectivity-filter-swapped mutants.
**Figure supplement 2.** Current-voltage relationship plots and representative current traces for the ramp pulse of CavMr-V6T selectivity-filter-swapped mutants.
**Figure supplement 3.** Current-voltage relationship plots and representative current traces for the ramp pulse of CavMr-G4S selectivity-filter-swapped mutants.
**Figure supplement 4.** Current-voltage relationship plots and representative current traces for the ramp pulse of NavPp-D4G selectivity-filter-swapped mutants.
**Figure supplement 5.** Current-voltage relationship plots and representative current traces for the ramp pulse of NavPp-T6V selectivity-filter-swapped mutants.

T6A 'TLEDWAD' showed no inward current in bath solutions containing divalent cations, suggesting that the $Ca^{2+}$-induced block was enforced (**Figure 6e**: bottom). Therefore, both of the selectivity filter sequences that provide $Ca^{2+}$ selectivity to canonical BacNavs failed to generate $Ca^{2+}$-permeable NavPp, indicating that the cation-permeable mechanism of NavPp differs from that of canonical BacNavs, as well as that of CavMr. On the other hand, the cells transfected with genes from *H. elongata* and *T. turnerae* failed to show any detectable currents, while these genes code selectivity filter sequences that are similar to that of NavPp (**Figure 1c**).

## Swapping the filter regions between CavMr and NavPp revealed the importance of the glycine residue at position 4 for $Ca^{2+}$-selective permeation

To search for the determinants of $Ca^{2+}$ selectivity in CavMr, we investigated a series of mutants in which the filter regions were swapped between CavMr and NavPp (**Figure 7a and b**), which exhibited channel activity (**Figure 7—figure supplements 1–2**). A NavPp mutant whose selectivity filter was replaced with that of CavMr, named NavPp-Mr, exhibited much higher $Ca^{2+}$ selectivity ($P_{Ca}/P_{Na} = 215 \pm 33$) as well as high $Sr^{2+}$ selectivity, comparable to that of CavMr (**Figure 7c**; **Figure 7—source data 1**). In addition, NavPp-Mr excluded $Na^+$ and $K^+$ similar to CavMr, but weakly allowed $Cs^+$ permeation in contrast to CavMr. On the other hand, a CavMr mutant whose selectivity filter

was replaced with that of NavPp (CavMr-Pp) almost lost its $Ca^{2+}$ selectivity ($P_{Ca}/P_{Na}$ = 13.8 ± 2.0), and was less able to discriminate $Cs^+$ and $K^+$ from $Na^+$ (**Figure 7d**; **Figure 7—source data 2**). That is, CavMr-Pp was a more non-selective channel than the wild-type CavMr, rather than a $Na^+$-selective channel. The $Ca^{2+}$ selectivity (from NavPp to CavMr) was almost transferable, but the $Na^+$ selectivity was not. We also investigated the full swapping of the filter sequences between CavMr and NavAb (**Figure 7a**), but neither swapped mutants of CavMr nor NavAb had detectable currents. This finding suggested that CavMr and NavAb achieve cation selectivity using different structural backbones and mechanisms.

Positions 4 and/or 6 of the filter sequences are thought to be important for $Ca^{2+}$-selective permeation through NavPp-Mr and CavMr, because only these two positions were mutated in the swapping experiments. We investigated which of the mutations in positions 4 and 6 had greater effects on the loss and acquisition of $Ca^{2+}$ selectivity in CavMr and NavPp, respectively. In CavMr, two single mutants, CavMr-G4D and CavMr-V6T, both decreased $Ca^{2+}$ selectivity and allowed $K^+$ and $Cs^+$ permeation (**Figure 8a;Table 1**; **Figure 8—figure supplements 1–2**; **Figure 8—source data 1**). The mutational effect was greater in CavMr-G4D, whose $P_{Ca}/P_{Na}$ was less than 10 (7.73 ± 2.24). CavMr-G4S, in which Gly4 was replaced with the Ser4 of NavAb, also exhibited lower $Ca^{2+}$ selectivity ($P_{Ca}/P_{Na}$ = 11.9 ± 1.5) and was also $K^+$ and $Cs^+$ permeable, indicating that a minor substitution by serine allowed the channel to retain a little calcium selectivity, but the monovalent cation selectivity had completely disappeared (**Figure 8b**; **Table 1**; **Figure 8—figure supplement 3**; **Figure 8—source data 2**). In the case of NavPp, NavPp-D4G acquired divalent cation, $Ca^{2+}$ and $Sr^{2+}$, selectivity over $Na^+$, and also showed a greater $P_K/P_{Na}$ and $P_{Cs}/P_{Na}$ than wild-type NavPp (**Figure 8c**; **Table 1**; **Figure 8—figure supplements 4–5**; **Figure 8—source data 3**). By contrast, NavPp-T6V failed to acquire the high $Ca^{2+}$ selectivity ($P_{Ca}/P_{Na}$ = 1.72 ± 1.09) and also allowed $K^+$ and $Cs^+$ permeation, while it had relatively high $Sr^{2+}$ selectivity. These results indicate that, in both CavMr and NavPp, a glycine residue at position 4 is a key determinant for $Ca^{2+}$ selectivity. It is noteworthy that the glycine is a conserved residue at position 4 of subdomains I and III in all subtypes of mammalian Cavs (**Figure 1b**).

## Discussion

### A native prokaryotic voltage-dependent $Ca^{2+}$ channel has a unique $Ca^{2+}$-selective mechanism

In this study, we newly characterized two prokaryotic voltage-dependent cation channels, CavMr and NavPp. CavMr is the first native prokaryotic Cavs reported, and NavPp could be inhibited by high concentrations of extracellular $Ca^{2+}$. The $P_{Ca}/P_{Na}$ of CavMr was more than 200 (**Figure 3e** and **Table 1**), comparable to that of CavAb, an artificial $Ca^{2+}$ channel. Anomalous mole fraction effects were not observed in CavMr (**Figure 4a and b**), suggesting that CavMr has a very high affinity for $Ca^{2+}$. In addition to providing new insights about general $Ca^{2+}$-selective mechanisms, CavMr has the potential to be a new genetic tool for upregulating calcium signaling, as BacNavs are useful genetic tools for increasing action potential firing in mice (**Bando et al., 2016**; **Kamiya et al., 2019**; **Lin et al., 2010**).

Phylogenetic analysis demonstrated that CavMr and NavPp are similar to each other, but distant from canonical BacNavs (**Figure 1b**). The high $Ca^{2+}$ selectivity of CavMr was transferable to NavPp. Intriguingly, two pairs of mutants with the same selectivity filter (CavMr-G4D and NavPp-T6V, CavMr-V6T and NavPp-D4G) showed a very similar tendency with regard to both the order and extent of cation selectivity (**Figure 8a and c**). Therefore, the basic overall architecture of the NavPp selectivity filter could be similar to that of CavMr. On the other hand, the $Ca^{2+}$-selectivity mechanism of CavMr completely differs from that of CavAb. Structural comparison of NavAb and CavAb showed that the aspartate mutations did not alter the main chain trace, and simply introduced the negative charges around the ion pathway to increase $Ca^{2+}$ permeability (**Figure 9a and b**) (**Tang et al., 2014**). By contrast, in the case of CavMr, two non-charged residues (Gly4 and Val6) are required for the high $Ca^{2+}$ selectivity (**Figures 7d** and **8a**), whereas Asp7 is not necessary (**Figure 4g**). A no-charge mutation at position 7, CavMr-D7M 'TLEGWVM', is an outstanding example demonstrating that high $Ca^{2+}$ selectivity can be achieved in the absence of any aspartates in its filter region. Furthermore, the introduction of a negative charge into the selectivity filter (G4D

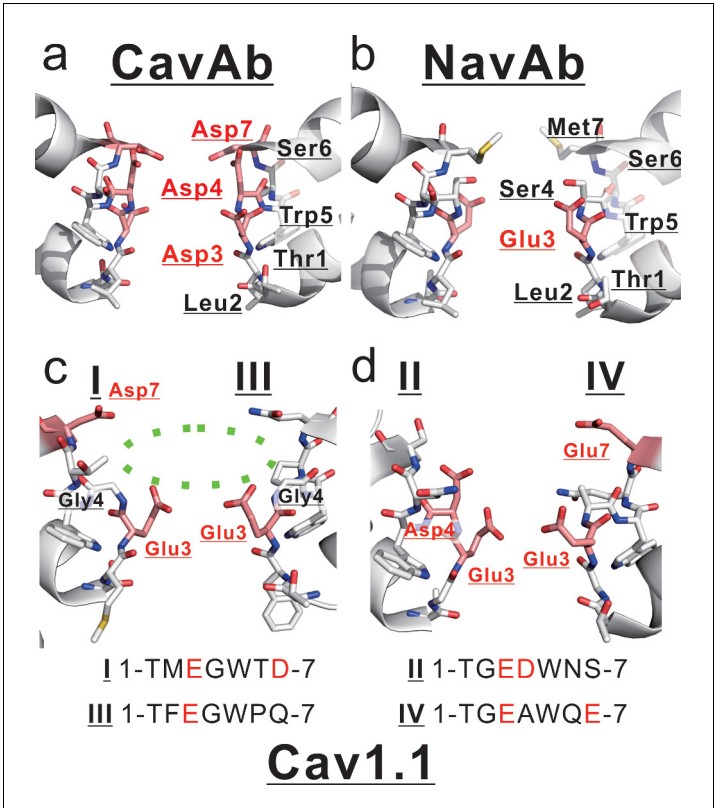

**Figure 9.** Comparison between mammalian and prokaryotic Cav. (**a, b**) Structures of the selectivity filter in CavAb (PDB code: 4MVZ) and NavAb (PDB code: 5YUA). (**c, d**) Structure of the rabbit Cav1.1 selectivity filter (PDB code: 5GJV). The subdomains I and III (**c**), and II and IV (**d**) are shown separately. The carbon atoms of negatively charged residues are indicated in pink. A dashed green circle indicates the wide entrance of the selectivity filter.

mutation) had an effect on the $Ca^{2+}$ selectivity of CavMr that was opposite to the effect seen in NavAb and NaChBac (*Tang et al., 2014*; *Yue et al., 2002*). Moreover, the decreased selectivity of monovalent and divalent cations in G4S also indicates that the glycine at position 4 plays a crucial role in $Ca^{2+}$ selectivity in CavMr (*Figure 8b*). The flexibility and/or small size of the glycine at position 4 in CavMr might be critical.

These findings are inconsistent with those derived from the $Ca^{2+}$-selective mutants of NavAb and NaChBac, and therefore the native structure of the selectivity filter and the molecular mechanism of ion selectivity of CavMr are thought to differ from those of CavAb. The structure of CavMr is not yet available, but we are able to speculate on the structure of the selectivity filter of CavMr, based on the structure of human Cav1.1 subdomains I and III (*Wu et al., 2016*) (*Figure 9c*), whose selectivity filter sequences are very similar to that of CavMr. In the selectivity filter of Cav1.1 subdomains I and III, the side chain of the residue at position 7 is shifted outward. The position 4 glycine residue widens the entrance of the selectivity filter, facilitating the entry of hydrated cations into the ion pore and possibly increasing $Ca^{2+}$ selectivity.

## The roles of Cavs in prokaryotes and the species-specific tuning of homo-tetrameric channels

Prokaryotes have a number of putative $Ca^{2+}$-binding proteins, such as EF-hand proteins, P-type $Ca^{2+}$ pumps, and $Ca^{2+}$ transporters (*Domínguez et al., 2015*). The intracellular $Ca^{2+}$ concentration is kept low and changes in response to mechanical and chemical stimuli (*Dominguez, 2004*). These features imply that prokaryotic $Ca^{2+}$ signaling is similar to that of eukaryotes. The strong ability of CavMr to exclude $Na^+$ and $K^+$ along with $Ca^{2+}$ permeation suggests that its primary physiological role is $Ca^{2+}$ intake in response to a voltage change (*Figure 3e and f*). In some bacteria, the direction of flagellar rotation and chemotaxis changes depending on the internal $Ca^{2+}$ concentration

(*Ordal, 1977*; *Tisa et al., 1993*; *Tisa and Adler, 1995*). *M. ruber* was isolated from hot springs, and therefore a sufficient amount of $Ca^{2+}$ is likely to exist in its native environment (*Loginova et al., 1984*). CavMr activation by a voltage change, which could vary depending on the environmental ionic conditions, might lead to any response that allows adaptation to the new environment, such as flagellar rotation. These characteristics indicate the existence of signal coupling between the membrane voltage and $Ca^{2+}$, even in the early stages of life, which might be the origin of the corresponding functions in eukaryotes, such as muscle contraction.

NavPp permeates more $Na^+$ than $Ca^{2+}$, but its selectivity is modest (*Figure 6d* and *Table 1*). Notably, *P. pacifica* is a marine myxobacterium that requires NaCl for its growth (*Iizuka et al., 2003*). As mentioned above, the basic architecture of the CavMr/NavPp group is thought to produce a preference for $Ca^{2+}$. *P. pacifica* might modify this channel architecture to acquire a $Na^+$ intake pathway, which would probably result in the remaining feature of low-affinity $Ca^{2+}$ inhibition in NavPp. This flexible usage of homo-tetrameric channels to allow different cations to permeate is also reported in another bacterium, *Bacillus alkalophilus* (*DeCaen et al., 2014*). NsvBa from *B. alkalophilus* is a non-selective channel whose selectivity filter is changed from 'TLESWAS', a typical $Na^+$-selective sequence in alkaliphilic bacillus, to 'TLDSWGS', possibly as an adaptation to its ionic environment. Recently, an early eukaryote, diatom, was found to have another homo-tetrameric channel with no selectivity, namely EukCat, which has an important role in electrical signaling in this species (*Helliwell et al., 2019*). These findings suggest that the cation selectivity of the homo-tetrameric channel family can be flexibly tuned to realize the required roles specific to its original species.

## Insights into $Ca^{2+}$ selectivity and the evolution of Cavs

Aspartate residues are generally observed in the $Ca^{2+}$ permeation pathway in ion channels, as well as in many $Ca^{2+}$-binding proteins (*Halling et al., 2016*; *Yan et al., 2015*; *Zalk et al., 2015*). Actually, NavAb and NaChBac were successfully transformed into $Ca^{2+}$-selective channels with the aspartate-introduced filter sequences 'TLDDW(S/A)D' (*Tang et al., 2014*; *Yue et al., 2002*). But, our results elucidate that this strategy is not the only way to achieve high $Ca^{2+}$ selectivity. Human Cav subdomains possess, at most, two aspartate residues in their selectivity filters in a part other than position 3. In addition, the negatively charged residue at position 3, which is thought to be the most critical for cation selectivity in both Navs and Cavs, is not aspartate but glutamate in most of the human Cav subdomains (*Yu and Catterall, 2004*). CavAb has 12 aspartates in the selectivity filter of its channel tetramer, while there are four aspartates in CavMr. The net negative charge is 5~7 in mammalian Cavs, 8 in CavMr, and 12 in CavAb. As shown in mammalian Cavs, $Ca^{2+}$ selectivity can be achieved with even fewer negative charges in the selectivity filter than is the case in CavAb, which suggest that the calcium-selective mechanism requires a specific backbone structure of the pore domain depending on its selectivity filter charges.

The members of the AnclNav group can be found in a variety of bacterial phyla and in archaea (*Figure 1—figure supplement 1*). In particular, the *Deinococcus-Thermus* phylum, in which *M. ruber* is included, is considered to be relatively close to the universal ancestor of life (*Hug et al., 2016*). It is also notable that the filter sequence of NavPp is completely the same as that of the ancestral Bac-Nav predicted in the previous report [TLED(or S in equal probability)WTD] (*Liebeskind et al., 2013*). These pieces of evidence suggest that the AnclNav group preserves the feature of an ancestral BacNav.

In our comprehensive phylogenetic analysis, all of the one-domain type of channel groups, including the AnclNav group, are almost equally distant from the root of the divided subdomains of eukaryotic 24TM Cavs/Navs (*Figure 1—figure supplement 2*). Again, this information is insufficient to allow us to deduce any conclusion for a eukaryotic ancestor of 24TM channels just before subdomain duplication. It is noteworthy, however, that the selectivity filter sequence of CavMr is very similar to those of human Cav subdomains I and III, both of which possess a glycine at position 4 (*Figure 9c*). In particular, the Cav3.1 and 3.2 subdomains I have the same sequence as the evolutionally distant CavMr. These sequence similarities of the glycine residue at position 4 are also found in CatSper, the sperm calcium permeable channels (*Darszon et al., 2011*), which branches close from the convergent point of four subdomains. The channel region of CatSper is formed by four different subunits (CatSper1–4). The selectivity filters of CatSper 3 'TVDGWTD' and CatSper 4 'TQDGWVD' are similar to that of CavMr. Taken together, these findings suggest that the selectivity filter of

eukaryotic ancestor of 24TM channels might have been similar to those of AnclNavs, especially to that of CavMr.

In the future, information about the structure of these homo-tetrameric channels could help us to gain a deeper understanding of channel evolution, and further investigation of the detailed structure of CavMr may help us to elucidate the principles and origin underlying $Ca^{2+}$ selectivity.

## Materials and methods

### Cloning of BacNav homologs and site-directed mutagenesis

The NaChBac amino acid sequence (NP_242367) was used as the query for a BLASTP search against the Microbial Genomic database at NCBI. The identified primary sequence data were obtained from Entrez at NCBI (*Meiothermus ruber* as ZP_04038264, *Plesiocystis pacifica* as ZP_01909854, *Halomonas elongata* as YP_003896792 and *Teredinibacter turnerae* as YP_003073405). These DNAs were synthesized by Genscript Inc and subcloned into the pCI vector (Promega) using the EcoRI and SalI sites and the pBiEX vector (Novagen) using the NcoI and BamHI sites, respectively. Site-directed mutagenesis was achieved by polymerase chain reaction (PCR) of the full-length plasmid containing the Nav gene using PrimeSTARMAX DNA Polymerase (Takara Bio.). All clones were confirmed by DNA sequencing.

### Phylogenetic analysis

Phylogenetic analyses were performed using the Molecular Evolutionary Genetics Analysis (MEGA X) software (*Kumar et al., 2016*). Protein sequences of putative BacNavs were collected by reference to a previous report (*Liebeskind et al., 2013*) and using multi-round searches of the NCBI database using NaChBac, NavAb, CavMr and NavPp as templates. Multiple sequence alignment was generated with MUSCLE contained in MEGA X. To generate the phylogenetic tree of BacNavs, the targeted sequences to be analyzed were selected using GBLOCKS0.91B (*Castresana, 2000*). The least stringent parameters selected 167 amino acids that cover most of transmembrane domains and the S4/S5 linker. For comparison with the eukaryotic Navs, Cavs, CatSper and EukCatA, the sequences of these proteins were collected by reference to the previous studies of Nav/Cav evolution (*Cai and Clapham, 2008*; *Gur Barzilai et al., 2012*; *Helliwell et al., 2019*; *Liebeskind et al., 2011*). The sequences of four-domain type of Navs and Cavs were divided to each subdomain. These divided subdomain sequences were aligned with EukCatA, CatSper and BacNavs using MUSCLE, and then the targeted sequences, which correspond to the selected amino acids in the analysis of BacNavs, were extracted. Regions that are poorly conserved between BacNavs and eukaryotic channels were manually removed, leaving 162 amino acids. Maximum likelihood trees were generated using MEGA X. The model validation was performed and determined to be a LG+G+F model in both the analyses, with and without eukaryotic channels. The sequence logos that indicate the frequencies at which different amino acids appear in the selectivity filter were generated with WebLogo (*Crooks et al., 2004*).

### Electrophysiological analysis using mammalian cells

For the recordings related to mole fraction effects (*Figure 4a–d*), currents were recorded from Chinese hamster ovary (CHO)-K1 cells (ATCC catalog number CCL-61) that expressed the channels. The recordings were performed as described previously (*Tateyama and Kubo, 2018*). Cells were transfected with channel DNAs using the LipofectAMINE 2000 (Invitrogen) and plated onto cover slips. Currents were recorded 24–36 hr after transfection. Current recording by the whole-cell patch-clamp technique was performed using Axopatch 200B amplifiers, Digidata1332A, and pClamp nine software (Molecular Devices). The pipette solution contained 130 mM KCl, 5 mM $Na_2$-ATP, 3 mM EGTA, 0.1 mM $CaCl_2$, 4 mM $MgCl_2$ and 10 mM HEPES (pH 7.2 adjusted with KOH). The bath solution contained 135 mM NaCl, 4 mM KCl, 1 mM $CaCl_2$, 5 mM $MgCl_2$ and 10 mM HEPES (pH 7.4 adjusted with NaOH). For the measurement of mole fraction effects, bath solutions containing different ratios of NaCl/$CaCl_2$ (135/0, 108/18, 81/36, 54/54, 27/82 and 0/90 mM) were used. A $Ca^{2+}$-free solution was achieved by using a solution containing 135 mM NaCl, 1 mM EGTA and 0 mM $CaCl_2$.

## Electrophysiological measurement in insect cells

Recordings other than those for mole fraction effects were performed using SF-9 cells. SF-9 cells (ATCC catalog number CRL-1711) were grown in Sf-900 III medium (Gibco) complemented with 0.5% 100 × antibiotic antimycotic (Gibco) at 27°C. Cells were transfected with target channel-cloned pBiEX vectors and enhanced green fluorescent protein (EGFP)-cloned pBiEX vectors using Fugene HD transfection reagent (Promega). The channel-cloned vector (2 µg) was mixed with 0.5 µg of the EGFP-cloned vector in 100 µL of the culture medium. Next, 3 µL Fugene HD reagent was added and the mixture was incubated for 10 min before the transfection mixture was gently dropped onto cultured cells. After incubation for 16–48 hr, the cells were used for electrophysiological measurements. In the measurement of I–V relation curves, the pipette solution contains 75 mM NaF, 40 mM CsF, 35 mM CsCl, 10 mM EGTA, and 10 mM HEPES (pH 7.4 adjusted by CsOH).

For evaluation of ion selectivity, a high-Na$^+$ pipette solution [115 mM NaF, 35 mM NaCl, 10 mM EGTA, and 10 mM HEPES (pH 7.4 adjusted by NaOH)] was used. For the evaluation of Ca$^{2+}$, Sr$^+$, K$^+$ and Cs$^+$ selectivity, Ca$^{2+}$ solution [100 mM CaCl$_2$, 10 mM HEPES (pH 7.4 adjusted by Ca(OH)$_2$), and 10 mM glucose], Sr$^{2+}$ solution [100 mM SrCl$_2$, 10 mM HEPES (pH 7.4 adjusted by Sr(OH)$_2$), and 10 mM glucose], K$^+$ solution [150 mM KCl, 2 mM CaCl$_2$, 10 mM HEPES (pH 7.4 adjusted by KOH), and 10 mM glucose], and Cs$^+$ solution [150 mM CsCl, 2 mM CaCl$_2$, 10 mM HEPES (pH 7.4 adjusted by CsOH), and 10 mM glucose], respectively, were used as the bath solution. $E_{rev}$ of highly Ca$^{2+}$-selective channels were measured under three external solutions containing: 144 mM NMDG-Cl and 4 mM CaCl$_2$; 135 mM NMDG-Cl and 10 mM CaCl$_2$; and 120 mM NMDG-Cl and 20 mM CaCl$_2$ (10 mM HEPES pH 7.4 adjusted with HCl). $E_{rev}$ of highly Ca$^{2+}$-selective channels for the calculation of $P_K/P_{Ca}$ and $P_{Cs}/P_{Ca}$ were measured under external solutions containing 135 mM NMDG-Cl and 10 mM CaCl$_2$ (10 mM HEPES pH 7.4 adjusted with HCl) with high-K$^+$ pipette solution [115 mM KF, 35 mM KCl, 10 mM EGTA, and 10 mM HEPES (pH 7.4 adjusted by KOH)] and high-Cs$^+$ pipette solution [115 mM CsF, 35 mM CsCl, 10 mM EGTA, and 10 mM HEPES (pH 7.4 adjusted by CsOH)], respectively. $E_{rev}$ of NavPp for the calculation of $P_{Ca}/P_{Na}$ or $P_{Sr}/P_{Na}$ were measured in an external solution containing 50 mM NMDG-Cl, 40 mM NaCl, 40 mM CaCl$_2$ or SrCl$_2$ and 10 mM HEPES (pH 7.4) adjusted with NaOH. $E_{rev}$ of NavPp for the calculation of $P_{Cs}/P_{Na}$ was measured using a high-Cs$^+$ pipette solution and an external solution containing 110 mM NMDG-Cl, 40 mM NaCl, 3 mM CaCl$_2$ and 10 mM HEPES (pH 7.4) adjusted with NaOH.

As the pipette solution for measurement of the Ca$^{2+}$ block effect in NavPp, low-Na$^+$ pipette solution [140 mM CsF, 10 mM NaCl, 10 mM EGTA, and 10 mM HEPES (pH 7.4 adjusted by CsOH)] and high-Na$^+$ pipette solution were used for inward and outward current measurement, respectively. As a bath solution, Ca$^{2+}$ blocking solution [30 mM NaCl, 120 mM NMDG-Cl, 1.5 mM CaCl$_2$, 10 mM HEPES (pH 7.4 adjusted by NaOH) and 10 mM glucose] was used for the 1.5 mM Ca$^{2+}$ blocking condition. In 10 mM Ca$^{2+}$ blocking condition, a bath solution contains 30 mM NaCl, 105 mM NMDG-Cl, 10 mM CaCl$_2$, 10 mM HEPES (pH 7.4 adjusted by NaOH) and 10 mM glucose. In each Ca$^{2+}$ blocking condition, 15 mM NMDG-Cl was replaced by 10 mM CaCl$_2$. Cancelation of the capacitance transients and leak subtraction was performed using a programmed P/10 protocol delivered at −140 mV. The bath solution was changed using the Dynaflow Resolve system. All experiments were conducted at 25 ± 2°C. Cells that have a leak current smaller than 0.5nA were used for measurement. When any outliers were encountered, these outliers were excluded if any abnormalities were found in other measurement environments, but were included if no abnormalities were found. All results are presented as mean ± standard error.

## Calculation of ion selectivity by the GHK equation

To determine the ion selectivity of each channel, the intracellular solution and extracellular solution were arbitrarily set and the reversal potential at each concentration was measured by giving the ramp pulse of membrane potential. The applied ramp pulse was set to include the reversal potential. In addition, a depolarization stimulus of 2–10 ms was inserted to check whether the behavior of the cell changed for each measurement. As a result, $P_{Ca}/P_{Na}$ was calculated by substituting the obtained reversal potential ($E_{rev}$) into the expression derived from the GHK equation (*Frazier et al., 2000*);

$$P_{Ca}/P_{Na} = \frac{-\left([\text{Na}^+]_{in} - [\text{Na}^+]_{out}e^{-E_{rev}F/RT}\right)\left(1 - e^{-2E_{rev}F/RT}\right)}{4\left([\text{Ca}^{2+}]_{in} - [\text{Ca}^{2+}]_{out}e^{-2E_{rev}F/RT}\right)\left(1 - e^{-E_{rev}F/RT}\right)}$$

where $F$ is Faraday's constant, $R$ is the Gas constant, and $T$ is 298.1 (K). The same expression was used for $Sr^{2+}$. The $Sr^{2+}$ selectivity ($P_{Sr}/P_{Na}$) was measured in the same way.

$Na^+$ selectivity against monovalent cations ($P_M/P_{Na}$) was calculated by substituting the obtained reversal potential and $P_{Ca}/P_{Na}$ into the expression derived from the GHK equation (*Lopin et al., 2012*):

$$P_{\mathrm{M}}/P_{\mathrm{Na}} = \left[ \frac{-4\left([Ca^{2+}]_{in}-[Ca^{2+}]_{out}e^{-2E_{rev}F/RT}\right)\left(1-e^{-E_{rev}F/RT}\right)}{\left([Na^+]_{in}-[Na^+]_{out}e^{-E_{rev}F/RT}\right)\left(1-e^{-2E_{rev}F/RT}\right)} \cdot (P_{Ca}/P_{Na}) - 1 \right] \left[ \frac{\left([Na^+]_{in}-[Na^+]_{out}e^{-E_{rev}F/RT}\right)}{\left([M^+]_{in}-[M^+]_{out}e^{-E_{rev}F/RT}\right)} \right]$$

## Acknowledgements

We appreciate the helpful suggestion by Dr. Yoshihiro Kubo about the structure of our manuscript. This work was supported by Grants-in-Aid for Scientific Research (S), a Grant-in-Aid for Young Scientists (B), the Japan Agency for Medical Research and Development, and the Toyoaki Scholarship Foundation.

# Additional information

## Competing interests
Yoshinori Fujiyoshi: is affiliated with CeSPIA Inc. The author has no other competing interests to declare. The other authors declare that no competing interests exist.

## Funding

| Funder | Grant reference number | Author |
| --- | --- | --- |
| Japan Agency for Medical Research and Development | | Yoshinori Fujiyoshi |
| Japan Science and Technology Agency | 15H05775 | Yoshinori Fujiyoshi |
| Japan Science and Technology Agency | 17K17795 | Katsumasa Irie |
| Toyoaki Scholarship Foundation | | Katsumasa Irie |

The funders had no role in study design, data collection and interpretation, or the decision to submit the work for publication.

## Author contributions
Takushi Shimomura, Conceptualization, Resources, Data curation, Formal analysis, Validation, Investigation, Writing - original draft, Writing - review and editing; Yoshiki Yonekawa, Data curation, Formal analysis, Validation, Writing - original draft; Hitoshi Nagura, Validation, Writing - original draft, Writing - review and editing; Michihiro Tateyama, Data curation, Formal analysis, Investigation; Yoshinori Fujiyoshi, Conceptualization, Funding acquisition, Validation, Writing - review and editing; Katsumasa Irie, Conceptualization, Data curation, Software, Formal analysis, Supervision, Funding acquisition, Validation, Investigation, Visualization, Methodology, Writing - original draft, Project administration, Writing - review and editing

## Author ORCIDs
Takushi Shimomura https://orcid.org/0000-0002-8109-535X
Katsumasa Irie https://orcid.org/0000-0002-8178-1552

## Decision letter and Author response
Decision letter https://doi.org/10.7554/eLife.52828.sa1
Author response https://doi.org/10.7554/eLife.52828.sa2

## Additional files

### Supplementary files
- Transparent reporting form

### Data availability
Protein sequence data are available in the NCBI protein database.

The following previously published datasets were used:

| Author(s) | Year | Dataset title | Dataset URL | Database and Identifier |
|---|---|---|---|---|
| Lucas S, Copeland A, Lapidus A, Glavina del Rio T, Dalin E, Tice H, Bruce D, Goodwin L, Pitluck S, Kyrpides N, Mavromatis K, Ivanova N, Markowitz V, Cheng J-F, Hugenholtz P, Woyke T, Wu D, Tindal B, Klenk H-P and Eisen JA | 2009 | Ion transport protein [Meiothermus ruber DSM 1279] | https://www.ncbi.nlm.nih.gov/protein/ZP_04038264.1?report=genpept | NCBI Protein, ZP_040 38264 |
| Shimkets L, Ferriera S, Johnson J, Kravitz S, Beeson K, Sutton G, Rogers Y-H, Friedman R, Frazier M and Venter JC | 2010 | K+ transporter, Kef-type [Plesiocystis pacifica SIR-1] | https://www.ncbi.nlm.nih.gov/protein/ZP_01909854.1?report=genpept | NCBI Protein, ZP_0 1909854 |

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
