## [Decision Letter]

**Acceptance summary:**

The manuscript by Shimomura et al. describes a new group of prokaryotic, single-domain, tetrameric Ca^2+^-permeable channels that can be regarded as ancestors to the four-domain modern Ca^2+^ channels. These new channels retain high selectivity for calcium and robust voltage-dependence. The authors have performed phylogenetic, electrophysiological and mutational analysis that nicely substantiate their claims. The authors have modified their manuscript according to the reviewers’ suggestions. These newly-described channels help broaden our understanding of the evolution of channels in general, also suggesting that complex, Ca^2+^ and voltage-dependent processes evolved early in life. These new channels will also help to understand the evolution of the mechanisms of calcium-selectivity in all Ca^2+^-permeable channels. Finally, the availability of a single-domain Ca^2+^ channel might be beneficial for structural studies, as has been shown with prokaryotic sodium channels.

**Decision letter after peer review:**

Thank you for submitting your article "A native prokaryotic voltage-dependent calcium channel with a novel selectivity filter sequence" for consideration by *eLife*. Your article has been reviewed by three peer reviewers, including Leon D Islas as the Reviewing Editor and Reviewer #1, and the evaluation has been overseen by Richard Aldrich as the Senior Editor. The following individual involved in review of your submission has agreed to reveal their identity: Steve Jones (Reviewer #3). The reviewers have discussed the reviews with one another and the Reviewing Editor has drafted this decision to help you prepare a revised submission.

Summary:

This is a highly interesting paper in which the authors report Ca^2+^-selective (and Ca^2+^-blocked) single homology domain voltage-gated channels in some bacteria that are distantly related to the well-characterized prokaryotic voltage-gated Na^+^ channels. A glycine residue in the selectivity filter (conserved in DI and DIII filters of eukaryotic Cavs) is required for Ca^2+^ selectivity and it is suggested that these new channels could represent the missing prokaryotic ancestor of eukaryotic 4-domain Cavs.

The data are very interesting and the conclusions are of high relevance, but all reviewers concur that there are technical issues with the selectivity experiments, especially regarding determination of the reversal potentials.

Essential revisions:

The authors apply an uncommon design for the voltage ramps used for this purpose, and in several figures it seems that the reversal potential cannot be measured. Please show I-V plots for each ramp, not only the raw currents, so that the reversal potential can be estimated by the reader. In many of the figures, emphasis is given to the voltage protocol and the currents are impossible to see. I-V plots are a better way to display these data. You can even obtain I-Vs from averaging the three ramps, properly scaling the time as voltage. Better yet, reversal potentials should be determined from I-V s obtained from tail current protocol.

Determination of reversal potentials depends also on the contribution of leak current and/or endogenous currents. Authors need to rule out contributions from these factors by carefully applying p/n or other type of subtraction to only measure currents from the expressed channels.

One of the major implications of this work is that these new prokaryotic channels could represent ancestors of eukaryotic Cav channels, and the authors imply that 4-domain eukaryotic Cav/Nav channels should have a single domain Ca^2+^-selective prokaryotic ancestor. Indeed, the prokaryotic ancestor of eukaryotic 4-domain channels has not been identified and is unlikely to be a prokaryotic voltage-gated Na^+^ channel, thought the paper is missing the key citation (Liebeskind et al., 2013). It is therefore surprising and disappointing that the manuscript does not conduct a wider evolutionary analysis to determine if these new prokaryotic channels could be the ancestors of eukaryotic 4-domain channels, and simply states that this is likely to be the case. While CavMR shows striking similarity to the selectivity filter sequence of eukaryotic Cavs, this sequence alone is not long enough to confirm a special evolutionary relationship. The authors should attempt to support these claims with phylogenetic analysis to show whether or not these channels could be ancestors of eukaryotic 4-domain channels.

The authors speculate that eukaryotic Ca channels evolved from a CavMr-like prokaryotic ancestor. However, it is not clear that such Ca channels are conserved among prokaryotes, since CavMr is the only known example. Yes, prokaryotes are ancestral to eukaryotes, but there has been plenty of evolution in both lineages since the last common ancestor. If CavMr-like channels are rare in prokaryotes, they may have evolved independently. That possibility should be discussed.

---

## [Author Response]

We really appreciate that the editors and reviewers considered our manuscript as meaningful. We are very happy that you recognized that CavMr is not only the firstly identified prokaryotic calcium channel but also a potential ancestor of 24TM cation channels. We also thank you for the constructive comments. We are very sorry for the lack of figures for presenting the reversal potential. In this revised manuscript, we willingly prepared new graphs generated from the current traces of ramp pulse to indicate the value of the reversal potential.

As following, we hope we could reply to all essential comments and suggestions for improving our manuscript.

Essential revisions:The authors apply an uncommon design for the voltage ramps used for this purpose, and in several figures it seems that the reversal potential cannot be measured. Please show I-V plots for each ramp, not only the raw currents, so that the reversal potential can be estimated by the reader. In many of the figures, emphasis is given to the voltage protocol and the currents are impossible to see. I-V plots are a better way to display these data. You can even obtain I-Vs from averaging the three ramps, properly scaling the time as voltage. Better yet, reversal potentials should be determined from I-V s obtained from tail current protocol.

Following your suggestion, we provided new figures of I-V relationships of the current generated by ramp pulses in Figure 3, Figure 4, Figure 6—figure supplement 1, Figure 7—figure supplement 1-2 and Figure 8—figure supplement 1-5. For the channels which have the long-lasting currents, the I-V relationship curves were created by averaging the currents generated by three ramp pulses. For the channels which have rapidly inactivating currents, such as CavMr-G4S, the curves were created by the currents generated by red-line ramp pulse.

Determination of reversal potentials depends also on the contribution of leak current and/or endogenous currents. Authors need to rule out contributions from these factors by carefully applying p/n or other type of subtraction to only measure currents from the expressed channels.

We are very sorry again for forgetting to describe about p/n or other type of subtraction. We added the following sentence in the electrophysiologic analysis section of the Materials and methods; “Cancellation of the capacitance transients and leak subtraction was performed using a programmed P/10 protocol delivered at -140 mV.”

One of the major implications of this work is that these new prokaryotic channels could represent ancestors of eukaryotic Cav channels, and the authors imply that 4-domain eukaryotic Cav/Nav channels should have a single domain Ca^2+^-selective prokaryotic ancestor. Indeed, the prokaryotic ancestor of eukaryotic 4-domain channels has not been identified and is unlikely to be a prokaryotic voltage-gated Na^+^ channel, thought the paper is missing the key citation (Liebeskind et al., 2013). It is therefore surprising and disappointing that the manuscript does not conduct a wider evolutionary analysis to determine if these new prokaryotic channels could be the ancestors of eukaryotic 4-domain channels, and simply states that this is likely to be the case. While CavMR shows striking similarity to the selectivity filter sequence of eukaryotic Cavs, this sequence alone is not long enough to confirm a special evolutionary relationship. The authors should attempt to support these claims with phylogenetic analysis to show whether or not these channels could be ancestors of eukaryotic 4-domain channels.

Encouraged by the reviewers’ suggestion, we performed the phylogenetic analyses that we believe as long as it is reliable and refrained from using the expertise methods easily to avoid drawing wrong conclusions, as we are not experts in this field. Although we cannot directly tell whether the CavMr/NavPp (AnclNavs) represent the ancestor of eukaryotic 4-domain channels only from the phylogenetic analyses themselves, we believe that they are more likely to represent it based on the results of the phylogenetic analyses with some indirect evidence.

We would like to notify you that, in the manuscript and the following explanations, we explicitly distinguish the ancestral BacNav, which is the origin of the prokaryotic Nav/Cav-type channels, and the “eukaryotic ancestor” of 4-domain type Nav/Cav, which has been located at the origin of subdomain duplications. In the following, we will discuss the relatedness of AnclNavs with the ancestral BacNav in the analyses with eukaryotic channels (1) and within bacterial layer (2), and then the supporting evidences from the related studies (3). We also discuss the possible relationship of AnclNavs to the eukaryotic ancestor of 4-domain type of Nav/Cav (4).

1) We used full length channels for analyses, with reference to the studies of Liebeskind et al., 2013, and Helliwell et al., 2019. To compare the 4-domain type of eukaryotic Nav/Cav with the single domain channels, we divided the 4 subdomains into respective sequences. We successfully reproduced the phylogenetic trees in which overall evolutional relationships between eukaryotic Nav/Cav, CatSper, EukCatA, and BacNavs were consistent with some previous reports. The results showed that the AnclNav group is placed evolutionally close to the canonical BacNav group over the convergence point of each subdomain of the 4-domain channels, which suggests that AnclNavs are a candidate of the ancestral BacNav as well as canonical BacNavs. We cannot assert clearly whether AnclNav is the closest to an ancestor among BacNavs from the tree of Figure 1—figure supplement 1, because of their closeness to each other.

2) We found many CavMr/NavPp-like channels, namely the members of AnclNav group, in the NCBI database. AnclNavs formed a distinct group from the canonical BacNavs, a NavAb-like and a *Bacillus* groups. Interestingly, the NavAb-like group belongs to only a few bacterial phyla, mainly proteobacteria, whereas AnclNavs belongs to various phyla. In particular, *Meiothermus ruber*, which holds CavMr, belongs to the Deinococcus-Thermus, a phylum that is believed to be closer to the evolutional origin than proteobacteria. These evidences suggest that AnclNavs are a more universal and widely distributed group and have deeper origin than canonical BacNavs.

3) The filter sequence of ancestral BacNav predicted by Liebeskind et al., 2013. is similar to that of AnclNavs, which is the reason why we named CavMr/NavPp group as so. In particular, the NavPp sequence is in complete agreement with the most likely sequence (TLED(or S in equal probability) WTD).

We replaced the phylogenetic tree in the previous Figure 1A to the new Figure 1B and added new two trees in the Figure 1—figure supplement 1 and 2. Based on these analyses with the supporting evidences, we added the description of revised phylogenetic analyses in the Results section and of the possibility that AnclNavs represent the ancestral BacNav in the Discussion section as follows;

(Results); “Phylogenetic analysis of these channel genes revealed… put AnclNavs closest to a *Bacillus* group (Figure 1—figure supplement 2).”

(Discussion); “The members of AnclNav group can be found…that the AnclNav group preserves the feature of an ancestral BacNav.”

4) In the phylogenetic tree including eukaryotes and prokaryotes (Figure 1—figure supplement 1-2), the convergent point of 4 subdomains of eukaryotic Nav/Cav is almost equally distant from the other 1-domain channels. Hence, we cannot identify which single-repeat channel preserves the most of the features of an ancestor of 4-domain Nav/Cav, based on our phylogenetic analysis. Nevertheless, the similar filter sequences (TxEGWxD) between two distant homologs, namely human Cav subdomains I/III and CavMr, implies that the eukaryotic ancestor of 4-domain Nav/Cav also had the similar features to CavMr, at least around the selectivity filter. Actually, the fact that CavMr is a Ca^2+^ selective channel enforces this implication. In addition, among the CatSper family that is close to the convergent point of 4 subdomains, CatSper3 and 4 have the filter sequences of TxDGWxD. Therefore, the prokaryotic ancestor of 4-domain channels might be also similar to AnclNavs, especially to CavMr. We added the description of this perspective in Discussion as follows;; “In our comprehensive phylogenetic analysis, … the selectivity filter of eukaryotic ancestor of 24TM channels might have been similar to those of AnclNavs, especially to that of CavMr.”

The authors speculate that eukaryotic Ca channels evolved from a CavMr-like prokaryotic ancestor. However, it is not clear that such Ca channels are conserved among prokaryotes, since CavMr is the only known example. Yes, prokaryotes are ancestral to eukaryotes, but there has been plenty of evolution in both lineages since the last common ancestor. If CavMr-like channels are rare in prokaryotes, they may have evolved independently. That possibility should be discussed.

As aforementioned, we searched for some AnclNav members, which are phylogenetically classified as a different branch from canonical BacNavs. In most cases, their filter sequences were either TLEGW (CavMr-type) or TLEDW (NavPp-type). The orthologues with the TLEGW sequences in AnclNavs were distributed in various phyla, such as plantomyces, cyanobacteria, Proteobacteria and so on, and not only in these Bacteria but also in Archaea. This result suggests that the TLEGW sequence is not the rare type of channels which have been evolved locally around *M. ruber*-related species, but rather, likely to be more general type than the canonical NavAb-like group. These descriptions and discussion of this point were included in the parts of manuscript above mentioned.